# TACTICS ON REFINING DECISION BOUNDARY FOR IMPROVING CERTIFICATION-BASED ROBUST TRAINING

## ABSTRACT

In certification-based robust training, existing methods utilize relaxation based methods to bound the worst case performance of neural networks given certain perturbation. However, these certification based methods treat all the examples equally regardless of their vulnerability and true adversarial distribution, limiting the model's potential in achieving optimal verifiable accuracy. In the paper, we propose new methods to include the customized weight distribution and automatic schedule tuning methods on the perturbation schedule. These methods are generally applicable to all the certifiable robust training with almost no additional computational cost. Our results show improvement on MNIST with $\epsilon = 0.3$ and CIFAR on $\epsilon = 8/255$ for both IBP and CROWN-IBP based methods.

## 1 INTRODUCTION

Deep neural networks (DNNs) have been shown to be highly vulnerable to adversarial attacks, carefully-crafted inputs that are nearly indistinguishable from naturally-occurring data but are misclassified by the network (Goodfellow et al., 2014; Szegedy et al., 2014). There exist many algorithms for both crafting adversarial attacks (Papernot et al., 2016) and building neural networks that are robust against such attacks. Fast gradient sign method (FGSM) (Goodfellow et al., 2014) was the very first approach to generate strong adversary. Projected gradient descent (PGD) (Madry et al., 2018) is one of the most successful and widely-used defense methods available. Adversarial training seeks to minimize the worst-case loss under adversarial perturbations within a pre-defined perturbation level, where multi-step PGD is used to estimate the worst-case attack during training. Compared to standard training, the adversarial term introduces risk of over-fitting (Moosavi-Dezfooli, 2021; Rice et al., 2020; Wang et al., 2019) and training instability (Tsipras et al., 2019; Zhang et al., 2020b) for adversarial training. There exist many related works on improving the model performance by additional regulation (Zhang et al., 2020b; Cisse et al., 2017) and customizing training curriculum (Zhang et al., 2019; Wang et al., 2020; Cai et al., 2018) for attack-based scenario. While adversarial training has been shown to be empirically effective against many types of attacks, it cannot be proven that the resultant models are robust against all adversaries. In fact, it has been shown that many defense methods dependent on heuristic techniques, including adversarial training, can be bypassed by stronger adversaries (Athalye et al., 2018).

This has motivated a separate branch of research focused on robustness certification/verification: computing provable guarantees on the robustness performance of neural networks against inputs with arbitrary perturbations within some $\ell_p$ norm-bounded ball. There are two main types of verification methods: complete and incomplete verification. The former computes exact robustness bounds using computationally-expensive methods such as mixed-integer programming (MIP) (Tjeng et al., 2019; Bunel et al., 2018), whereas the latter provides looser robustness bounds with different branches of methods like randomized smoothing (Cohen et al., 2019; Salman et al., 2019; Lecuyer et al., 2019), Lipschitz-based robustness(Trockman & Kolter, 2021; Tsuzuku et al., 2018) and convex adversarial polytype (Weng et al., 2018; Zhang et al., 2018; Dvijotham et al., 2018b; Gowal et al., 2019) on which in this paper we mainly focus. A sparsely related branch of research, called certified robust training (Dvijotham et al., 2018a; Gowal et al., 2019; Zhang et al., 2020a), aims to train a certifiably robust model. These methods compute verified robustness bounds and incorporate

them into the training process, such that the resultant models can then be proven to be robust using a verification method.

Currently, the most efficient certified training method is interval bound propagation (IBP) (Gowal et al., 2019), which requires only two additional forward passes during training. It is important to note that IBP is significantly more efficient than adversarial training, which often require many PGD steps to achieve high robustness (Madry et al., 2018). In standard certified training, a uniform $\epsilon$ and loss function weight are usually used across all the training examples, this is not ideal for optimal verified accuracy since the examples are not necessarily equivalent in terms of vulnerability.

## 1.1 OUR CONTRIBUTIONS

In this paper, we propose two novel methods to improve robustness via refining certified decision boundary for verifiable adversarial training methods such as IBP and CROWN-IBP. In particular, our algorithms are generally applicable to all the verifiable adversarial training methods at almost no additional computational cost. We sum up the key contributions as following: 1. Zeng et al. (2020) pointed out the adversarial example distribution deviates from the clean data. We further analyze the importance weight to correct empirical distribution from a novel perspective and theoretically prove more weight is needed for the examples closer to decision boundary if sampled from clean data distribution. 2. Building upon previous analysis, we come up with a symmetrical re-weighting function based on worst case margin of the correct labels, emphasizing the examples around the decision boundary. 3. For verifiable adversarial perturbation, empirically a slightly larger $\epsilon_{train}$ achieves optimal evaluation robustness accuracy on $\epsilon_{eval}$, while this uniform setup is not ideal for examples around decision boundary. To address the issue of large perturbation, we developed an auto-tuning algorithm to customize the $\epsilon_{train}$ for each individual example.

## 2 BACKGROUND

### 2.1 ADVERSARIAL ATTACKS AND TRAINING

Let $\mathcal{D} = \{(\boldsymbol{x}_i, y_i)\}_{i=1}^n$ represent the dataset, where $\boldsymbol{x}_i \in \mathcal{X}$ and $y_i \in \mathcal{Y} = \{0, 1, ..., C-1\}$. Let $\mathcal{B}(\boldsymbol{x}_i, \epsilon)$ denote the set of points in the $\ell_p$-norm ball with radius $\epsilon$ around $\boldsymbol{x}_i$. The objective function is

$$\min_{\theta} \frac{1}{n} \sum_{i=1}^n l(f_\theta(\boldsymbol{x}_i'), y_i), \text{ where } \boldsymbol{x}_i' = \underset{\boldsymbol{x}' \in \mathcal{B}(\boldsymbol{x}_i, \epsilon)}{\arg \max} \, l(f_\theta(\boldsymbol{x}'), y_i) \tag{1}$$

$f_\theta : \mathcal{X} \to \mathbb{R}^C$ is a score function and $l : \mathbb{R}^C \times \mathcal{Y} \to \mathbb{R}$ is the loss function. Adversarial training tackles this min-max problem by alternating between solving inner loop by attack method to estimate the worst-case adversarial scenario in $\mathcal{B}(\boldsymbol{x}_i, \epsilon)$ and outer loop to update the model parameters and minimize loss function.

### 2.2 CERTIFIED TRAINING

**Robustness Verification.** Different from attack-based training, certified training provides a guarantee for worst case scenario via bounding the neural network outputs. The certified accuracy is the lower bound of robustness accuracy under any attack method, thus improving the certified accuracy helps in understanding the potential of the neural network in defending against adversarial attacks.

**Interval Bound Propagation.** There have been many proposed neural network verification algorithms that can be used to compute worst case output. IBP (Gowal et al., 2019) utilizes a simple, interval arithmetic approach to propagate bounds through a network. Let $\boldsymbol{z}_{k-1}$ denote the input to a layer $\boldsymbol{z}_k$. IBP bounds $\boldsymbol{z}_k$ by computing lower and upper bounds $\underline{\boldsymbol{z}}_k, \overline{\boldsymbol{z}}_k$ such that $\underline{\boldsymbol{z}}_k \leq \boldsymbol{z}_k \leq \overline{\boldsymbol{z}}_k$ holds element-wise. For affine layers represented by $h_k(\boldsymbol{z}_{k-1}) = W\boldsymbol{z}_{k-1} + b$, IBP computes: $\underline{\boldsymbol{z}}_k = W \frac{\overline{\boldsymbol{z}}_{k-1} + \underline{\boldsymbol{z}}_{k-1}}{2} - |W| \frac{\overline{\boldsymbol{z}}_{k-1} - \underline{\boldsymbol{z}}_{k-1}}{2} + b$ and $\overline{\boldsymbol{z}}_k = W \frac{\overline{\boldsymbol{z}}_{k-1} + \underline{\boldsymbol{z}}_{k-1}}{2} + |W| \frac{\overline{\boldsymbol{z}}_{k-1} - \underline{\boldsymbol{z}}_{k-1}}{2} + b$. Propagating the bounds through the network allows us to compute the upper and lower bound of last layer logits $\overline{\boldsymbol{z}}_K, \underline{\boldsymbol{z}}_K$ and evaluate if an input $\boldsymbol{x}$ is verifiably robust. The logit of the true class equals the lower bound and the logits of other classes equal to the upper bound:

$$\hat{z}_{K,m} = \begin{cases} \underline{z}_{K,m} & \text{if } m \text{ is the true class} \\ \overline{z}_{K,m} & \text{otherwise} \end{cases} \tag{2}$$

IBP training uses a hyperparameter schedule on $\epsilon$ (starting from 0 and increasing to $\epsilon_{train}$, typically set at $\epsilon_{train}$ which is slightly larger than $\epsilon_{eval}$) and a mixed loss function that combines natural and robust cross-entropy loss: $\min_\theta E_{(\boldsymbol{x},y)\sim\mathcal{P}}[\kappa l(\boldsymbol{z}_K, y) + (1-\kappa)l(\hat{\boldsymbol{z}}_K, y)]$, where $\mathcal{P}$ is the data distribution, $l$ is the cross entropy loss, $\kappa$ is a hyperparameter that balances the weight between natural and robust loss, and $\hat{\boldsymbol{z}}_K$ represents the worst case logits computed using IBP.

**CROWN-IBP.** CROWN was introduced by Zhang et al. (2018) and achieves a tight bound by adaptively selecting the linear approximation. Zhang et al. (2020a) proposed CROWN-IBP combining IBP forward bounding pass and CROWN style backward bounding pass. CROWN-IBP trained models has a tighter bound compared with IBP models under the IBP metric, with incurring cost on computational efficiency from CROWN backward propagation and generally requiring more epochs for training stability.

### 2.3 ATTACK-BASED RE-WEIGHTING

Motivated by the idea that all data points are not equally vulnerable to adversarial attack, researchers proposed methods to re-weight the minimax risk by adding a re-weighting term $\omega(\boldsymbol{x}_i, y_i)$ ahead of individual example loss. For instance, Zeng et al. (2020) noted the adversarial distribution deviation from the clean examples and assigned weights that monotonically decreases with the examples' confidence margin. The "confidence margin" is calculated by attack methods such as PGD, then the risk is re-weighted by a parameterized exponential family as: $\min_\theta \frac{1}{n}\sum_{i=1}^n \omega(\boldsymbol{x}_i, y_i)l(f_\theta(\boldsymbol{x}_i'), y_i)$, s.t. $\omega(\boldsymbol{x}_i, y_i) = \exp(-\alpha \operatorname{margin}(f_\theta, \boldsymbol{x}_i + \delta_i, y_i))$, where $\alpha$ is a positive hyper-parameter and the new risk biases larger weights towards the mis-classified examples.

Zhang et al. (2020c) propose GAIRAT (Geometry-Aware Adversarial Training), a method to reweight adversarial examples based on how close they are to the decision boundary. During the training process, GAIRAT explicitly assigns larger/smaller weights to data points closer/farther to the decision boundary respectively: $\omega(\boldsymbol{x}_i, y_i) = (1 + tanh(\lambda + 5 \times (1 - 2 \times \kappa(\boldsymbol{x}_i, y_i)/K)/2$, where $\lambda$ is a hyper-parameter, $K$ is the maximal allowed attack iteration and $\kappa$ is the least iteration number that the attack method requires to fool the classifier. Similar to re-weighting, Wang et al. (2021) improves clean image performance by prioritizing the robustness between the most dissimilar groups

### 2.4 CUSTOMIZED ADVERSARIAL TRAINING

In most adversarial training methods, the adversarial attack strength usually follows a pre-defined scheduler throughout the training process. For instance, the perturbation $\epsilon$ is a uniform number for all examples and usually gradually increases. Cheng et al. (2020) argued that this assumption may be problematic given the fact that adversarial examples are not equally vulnerable and proposed an auto-tuning $\epsilon$ method by assigning individual $\epsilon_i$ to each data and increasing it if the current attack is successful. In Zhang et al. (2020b), they proposed friendly adversarial training (FAT) by progressively increasing the attack perturbation with early-stop PGD and alleviating the issue of strong adversarial attack and cross-over mixture.

## 3 OUR PROPOSED METHODS

### 3.1 BOUND-BASED WEIGHTED LOSS

In classical classification tasks, training minimizes the following loss function: $\mathbb{E}_{(\boldsymbol{x},y)\sim P}[l(f_{\boldsymbol{\theta}}(\boldsymbol{x}), y)]$ estimated by $\frac{1}{n}\sum_{i=1}^n [l(f_{\boldsymbol{\theta}}(\boldsymbol{x}_i), y_i)]$. For adversarial training, let $\boldsymbol{x}'$ be the worst case input under verifiable adversary or perturbed example from attack, the adversarial examples $(\boldsymbol{x}', y) \sim P'$, where $P'$ is an unknown distribution dependent on the clean example distribution $P$, perturbation method and neural network parameters. In practice, the training objective is usually $\frac{1}{n}\sum_{i=1}^n [l(f_{\boldsymbol{\theta}}(\boldsymbol{x}_i'), y_i)]$ estimating $\mathbb{E}_{(\boldsymbol{x},y)\sim P}[l(f_{\boldsymbol{\theta}}(\boldsymbol{x}'), y)]$, while the true objective should

be $\mathbb{E}_{(\boldsymbol{x}',y)\sim P'}[l(f_{\boldsymbol{\theta}}(\boldsymbol{x}'),y)]$. There exists a discrepancy between the true distribution and empirical sampling method. To bridge this gap, we follow Zeng et al. (2020) and introduce importance weight

$$s(f_{\boldsymbol{\theta}},\boldsymbol{x}_i',y_i) := \frac{P'(\boldsymbol{x}_i',y_i)}{P(\boldsymbol{x}_i,y_i)} \tag{3}$$

s.t.

$$\frac{1}{n}\sum_{i=1}^{n}[l(f_{\boldsymbol{\theta}}(\boldsymbol{x}_i'),y_i)s(f_{\boldsymbol{\theta}},\boldsymbol{x}_i',y_i)] \approx \mathbb{E}_{(\boldsymbol{x},y)\sim P}[l(f_{\boldsymbol{\theta}}(\boldsymbol{x}'),y)\frac{P'(\boldsymbol{x}',y)}{P(\boldsymbol{x},y)}] \tag{4}$$

$$\approx \mathbb{E}_{(\boldsymbol{x}',y)\sim P}[l(f_{\boldsymbol{\theta}}(\boldsymbol{x}'),y)]$$

As further exploration, we analyze the importance weight:

$$\frac{P'(\boldsymbol{x}_i',y_i)}{P(\boldsymbol{x}_i,y_i)} = \frac{P'(\boldsymbol{x}_i'|y_i)P'(y_i)}{P(\boldsymbol{x}_i|y_i)P(y_i)} = \frac{P'(\boldsymbol{x}_i'|y_i)}{P(\boldsymbol{x}_i|y_i)} \tag{5}$$

where the first equality follows from the definition of conditional probability and second equality is from the fact that the label $y_i$ and its distribution does not change through the adversarial process.

The probability ratio gives us more insight in designing the parametric function for importance weight. Given label $y_i$, there exists a distribution of its corresponding clean image space $P(\boldsymbol{x}|y_i)$ and similarly for worst case input space $P'(\boldsymbol{x}'|y_i)$. If the distribution of its corresponding worst case $\boldsymbol{x}_i'$ gets more concentrated compared to clean image distribution $P(\boldsymbol{x}|y_i)$, then more weight should be given to this example and vice versa.

**Theorem 1.** *Given a binary classification task $\mathbb{R}^n \to \{+,-\}$ with equal prior probability, assume the corresponding examples $\boldsymbol{x}_+$ and $\boldsymbol{x}_-$ are uniformly distributed in region $\mathbb{S}_+, \mathbb{S}_- \subset \mathbb{R}^n, \mathbb{S}_+ \cap \mathbb{S}_- = \emptyset$. If there exists a bijection mapping $m := \mathbb{S}_+ \mapsto \mathbb{S}_-, \mathbb{S}_- \mapsto \mathbb{S}_+$ s.t. the post-mapping example $\boldsymbol{x}_+'$ and $\boldsymbol{x}_-'$ distribution remains uniform in each region, the expectation of conditional distribution ratio $\frac{P'(\boldsymbol{x}'|y)}{P(\boldsymbol{x}|y)}$ over original distribution $P(\boldsymbol{x},y)$ is great or equal than 1.*

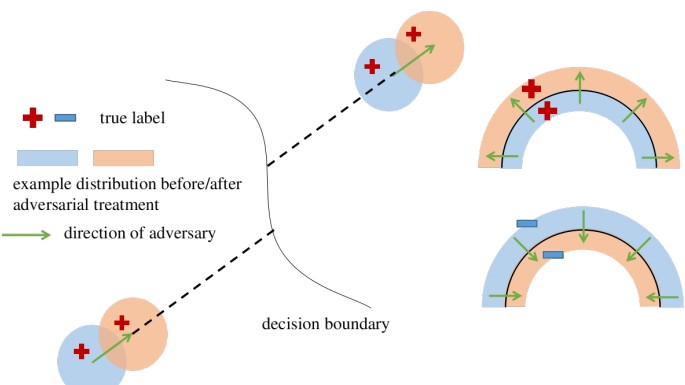

Figure 1: Example distribution movement: far away from decision boundary(left) vs close to decision boundary(right)

**Example 1.** Figure 1 shows two extreme cases of adversarial example $\boldsymbol{x}'$ distribution movement from clean $\boldsymbol{x}$ distribution given decision boundary in a simplified 2D scenario. When $\boldsymbol{x}$ is far away from decision boundary, the direction of attack or movement towards worst case example under verifiable adversary is same for all the $\boldsymbol{x}$. In this case, $\frac{P'(\boldsymbol{x}'|y)}{P(\boldsymbol{x}|y)} \approx 1$. In the second scenario, assume the decision boundary is a perfect semicircle representing the local curvature of decision boundary and perturbed examples from both classes follow a uniform distribution, then the examples with $'-'$ label in the outer arc condensed into the inner arc and vice versa for examples with $'+'$ label moving to the external arc. Due to the existence of curvature, the area of the outer arc $S_{in}$ is

greater than the inner arc $S_{out}$. Therefore, from Theorem 1 the expectation of $\frac{P'(\boldsymbol{x}'|y)}{P(\boldsymbol{x}|y)}$ is greater than 1. Another detailed example is illustrated in Appendix B. In this later example, the uniform perturbation assumption replaced the post-mapping uniform distribution assumption, which is more realistic for adversarial scenario.

From the two extreme case analysis, we claim that the importance weight around decision boundary should be larger due to the local curvature, since any arbitrary $\boldsymbol{x}'$ and $\boldsymbol{x}$ distribution pairs (local curvature is relatively larger than perturbation distance but its effect is not negligible on post adversarial distribution) can be seen as an intermediate case of two extremes. Therefore, we design a parametric weight function emphasizing the examples around the decision boundary in a symmetrical manner.

Similar to Zhang & Liang (2019), we define the margin of classifier $f$ for a data point $(\boldsymbol{x}, y)$ as the probability difference between correct label and the most confident label among others. Specially, we use bound-based method such as IBP to get the margin under perturbation: $\hat{P} = \text{softmax}(\hat{\boldsymbol{z}}_K)$, where $\hat{\boldsymbol{z}}_K$ is the worst case logits as defined in equation 2, $\hat{P}$ is the probability distribution calculated by softmax function given $\hat{\boldsymbol{z}}_K$.

$$\text{margin}(f, \boldsymbol{x}, y) = \hat{P}(f(\boldsymbol{x}) = y) - \max_{m \neq y} \hat{P}(f(\boldsymbol{x}) = m) \tag{6}$$

The margin is positive when the example can be certified robust, otherwise the margin is negative. The magnitude of margin heuristically indicates the "distance" of the example towards the current decision boundary. Left half of Figure 2 shows our implementation of re-weighting against vulnerability of examples, where the size of the labels indicate the weights and corresponding perturbations on clean data points. The most vulnerable examples are around the decision boundary, since the worst case scenario of a small perturbation could cross the decision boundary and fool the classifier. For the perturbed data points with smaller margin absolute value (closer to boundary), despite it being correctly classified, we assign larger weights to help the model focus on these data and accommodate more slightly mis-classfied data. On the other hand, for the data points farther from decision boundary, if it is mis-classfied (large negative margin), there is no need to assign large weight since the model capacity is limited. For correctly classified data, we also assign small weights because they are less related to the decision boundary.

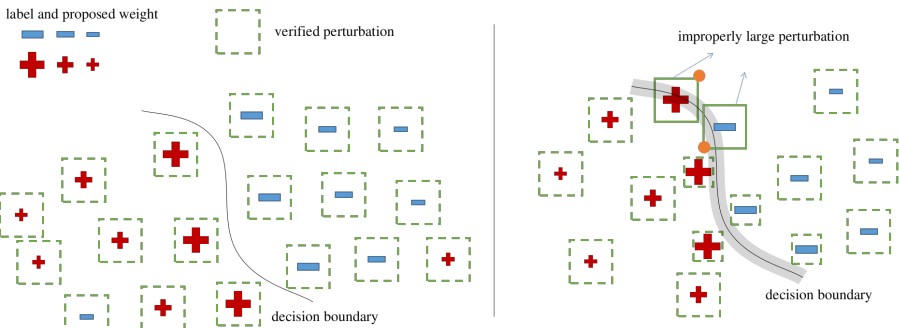

Figure 2: Left: Re-weighting examples based on vulnerability, Right: Auto-tuning $\epsilon$: customize the perturbation for mis-classified examples

We parameterize the importance weight in the following form:

$$\omega_i = e^{-\gamma * (|\text{margin}(f_{\boldsymbol{\theta}}, \boldsymbol{x}'_i, y_i, \epsilon)|)} + \alpha \sim s(f_{\boldsymbol{\theta}}, \boldsymbol{x}'_i, y_i) \tag{7}$$

where $\gamma$ and $\alpha$ are both positive hyper-parameters to balance the effect of re-weighting. The full loss function after re-weighting is then defined as:

$$\kappa \frac{1}{n} \sum_{i=1}^{n} l(\boldsymbol{z}_K, y) + (1 - \kappa) \frac{1}{\sum \omega_i} \sum_{i=1}^{n} \omega_i l(\hat{\boldsymbol{z}}_K, y) \tag{8}$$

where $\kappa$ is a hyper-parameter to balance the trade-off between clean loss and re-weighted robustness loss.

As comparison with other re-weighting techniques, Zhang et al. (2020c) gives larger weights to the vulnerable points defined by least attack iterations to fool the model. Zeng et al. (2020) parameterizes the weight with a similar exponential function given attack-based margin. Both of their heuristic weight functions monotonically decrease against margin, in which case a heavily mis-classified point shares a large weight and a well classified point has a smaller weight. Our bound-based method method, backed up by a qualitatively theoretical analysis, emphasizes the data points symmetrically around the decision boundary.

## 3.2 AUTO-TUNING $\epsilon$

In certified training, the training perturbation magnitude $\epsilon_{train}$ is usually larger than the $\epsilon_{eval}$ for optimal test accuracy. As Table 1 shows the robustness error against different $\epsilon_{train}$ and $\epsilon_{eval}$, the optimal testing accuracy for $\epsilon_{test} = 0.3, 0.35$ were both achieved at $\epsilon_{train} = 0.4$ while smaller $\epsilon_{train}$ fails certifying larger $\epsilon_{test}$.

Table 1: IBP certified robust error on MNIST data

|  | $\epsilon_{\text{test}} = 0.30$ | $\epsilon_{\text{test}} = 0.35$ | $\epsilon_{\text{test}} = 0.40$ | $\epsilon_{\text{test}} = 0.45$ |
|---|---|---|---|---|
| $\epsilon_{\text{train}} = 0.30$ | 9.81% | 100% | 100% | 100% |
| $\epsilon_{\text{train}} = 0.35$ | 8.76% | 12.13% | 100% | 100% |
| $\epsilon_{\text{train}} = 0.40$ | 8.66% | 11.40% | 15.82% | 100% |
| $\epsilon_{\text{train}} = 0.45$ (training gets unstable) | 25.13% | 30.62% | 37.63% | 47.88% |

With slightly larger perturbation during training, the model is more capable to handle unseen data in the testing. This observation is empirical, however, due to the limitation of model capacity, it is difficult for the model to handle all the examples with very large perturbation. When $\epsilon_{train} = 0.45$, we notice the training instability between different random seeds. Our intuitive explanation is shown in the right half of Figure 2: With large perturbation, for the vulnerable points around decision boundary, the worst case prediction (indicated by orange dots) protrudes the ideal boundary with a large margin and it is sometimes impossible for the model to fit these points. Including these worst case scenarios in training encourages over-fitting and compromises the model's ability to find the right boundary. In implementation, very large perturbation does not help improve the verified accuracy. Besides, during the training stage, a fast growing perturbation schedule may potentially cause instability.

Following the above discussion, a natural solution would be customizing perturbation for each example in the training set. For the majority of points in the interior of decision zone, we would like to encourage large perturbation for the model to enforce a thick decision "gap" (grey colored thick line) for robustness consideration. Ideally, we would like to have the interior points to see large perturbations for optimal robustness and boundary points with moderate perturbations to avoid over-fitting. In the following, we propose an adaptive procedure to customize the perturbation for each example based on verified bound. The algorithm is shown in Algorithm 1.

## 4 EXPERIMENTS

### 4.1 DATASETS AND IMPLEMENTATION

We directly use the code from (Zhang et al., 2020a) provided for IBP and CROWN-IBP and leverage the same CNN architecture(DM-Large) and training scheduler on MNIST and CIFAR-10 datasets.

For MNIST IBP schedule, the neural network is trained for 100 epochs with a batch size of 100. The base scheduler of training $\epsilon$ starts from 0 after 3 warm-up epochs and linearly grows to desired maximum training $\epsilon$ with 17 ramp-up epochs, after which base $\epsilon$ stays at the targeted level. The Adam optimizer learning rate is initialized at 0.001 and decays by 10 times after 25 and 42 epochs. With MNIST CROWN-IBP schedule, the model is trained for 200 epochs with a batch size of 256. The ramp-up stage starts from epoch 10 and ends at epoch 50.

For CIFAR-10 dataset, a total of 3200 epochs including 320 warm-up epochs is used for training with a batch size of 1024 and learning rate of 0.0005 following two $10\times$ decay after epoch 2600 and 3040. As a standard treatment, random horizontal flip and crops are used as data augmentation.

---

**Algorithm 1:** Bound-based Auto-Tuning algorithm

---

**Input:** Training data set $(X, Y)$, standard perturbation scheduler, perturbation bound method,
    auto-eps maximum perturbation offset $\epsilon_{maxoff}$, network parameters $\theta$, learning rate $\eta$
initialize perturbation offset $\epsilon_{i,off} = 0$ for all the examples, initialize $\theta$
**for** *epoch = 1,...,N* **do**
  **for** *batch = 1,...,M* **do**
    $\epsilon_{base} \leftarrow scheduler()$
    **for** *i = 1,...,B* **do**
      $\epsilon_i \leftarrow \epsilon_{base} - \epsilon_{i,off}$
      get the upper and lower bound of classification logits with bound method:
      $z(\hat{\epsilon_i}) = \{\overline{z(\epsilon_i)}, \underline{z(\epsilon_i)}\} \leftarrow \text{bound}(f_\theta, \boldsymbol{x}_i, y_i, \epsilon_i)$
      calculate the softmax worst case probability: $\hat{P} = \text{softmax}(\hat{z}(\epsilon_i))$
      calculate the margin: $\text{margin}_i = \hat{P}(y_i) - \max_{m \neq y_i} \hat{P}(m)$
      **if** *margin$_i$ < 0* **then**
        $\epsilon_{i,off} \leftarrow -\text{margin}_i \times \epsilon_{i,maxoff}$
      **else**
        $\epsilon_{i,off} \leftarrow 0$
      **end**
    **end**
    update the network parameter: $\theta \leftarrow \theta - \eta \nabla_\theta \sum_{i=1}^{B} l(f_\theta(\boldsymbol{x}_i + \epsilon_i, y_i))/B$
  **end**
**end**

---

Notice for CROWN-IBP implementation, we reduce the training batch size from 1024 to 256 due to memory constraint.

The hyper-parameter $\kappa$ defined in equation 8 linearly decreases from 1 to 0.5 during ramp-up stage. According to Zhang et al. (2020a), this scheduler ($\kappa_{start} = 1, \kappa_{end} = 0.5$) achieves better clean accuracy than ($\kappa_{start} = 0, \kappa_{end} = 0$) and ($\kappa_{start} = 0, \kappa_{end} = 0$) by ending up with weight on natural cross entropy loss. All the experiments were performed on 3 random seeds for reproducibility.

Our methods are generally applicable to any certifiable adversarial training method. In the next section, we first leverage IBP for illustrative examples followed by CROWN-IBP results.

### 4.2 RESULTS

#### 4.2.1 EFFECTS OF RE-WEIGHTING

In Table 2, we show the effect of hyper-parameters used for the re-weighting under IBP, where a larger $\gamma$ indicates a stronger re-distribution of weight for examples and $\alpha$ is a small number to prevent weight vanish when |margin| is close to 1. The hyper-parameters are evaluated on CIFAR-10 at $\epsilon = 8/255$ and MNIST at $\epsilon = 0.4$. Our optimal hyper-parameter turns out to be $\gamma = 5, \alpha = 0.1$ for both data sets. Re-weighting method gains $0.24\%$ verified accuracy on MNIST and $0.82\%$ on CIFAR-10.

The goal of re-weighting is to approximate the true adversarial distribution by sampling from clean data distribution. Unfortunately, it is difficult to visualize the intractable distribution. From another perspective, we can interpret re-weighting as concentration on the robustness decision boundary and encouragement for larger margin. Figure 3 shows the comparison of margin distribution for CIFAR-10 testing data set under testing $\epsilon = 8.8/255$. With the vanilla model, the example margins

Table 2: Robustness against different hyper-parameters with re-weighting with IBP

| *Dataset* | $\epsilon_{eval}$ | $\epsilon_{train}$ | $\gamma$ | $\alpha$ | Clean err.(%) | Verified err.(%) |
|-----------|-------------------|--------------------|----------|----------|---------------|------------------|
| *MNIST* | 0.3 | 0.4 | baseline | | $2.28 \pm 0.12$ | $8.66 \pm 0.05$ |
| | | | 1 | 0.1 | $2.28 \pm 0.13$ | $8.68 \pm 0.06$ |
| | | | 5 | 0.1 | $2.30 \pm 0.07$ | $\mathbf{8.42} \pm 0.07$ |
| | | | 10 | 0.1 | $2.28 \pm 0.05$ | $8.61 \pm 0.05$ |
| *CIFAR-10* | 8/255 | 8.8/255 | baseline | | $49.44 \pm 0.49$ | $72.26 \pm 0.16$ |
| | | | 1 | 0.1 | $49.06 \pm 0.01$ | $72.32 \pm 0.03$ |
| | | | 5 | 0.1 | $49.46 \pm 0.27$ | $\mathbf{71.44} \pm 0.04$ |
| | | | 10 | 0.1 | $50.11 \pm 0.38$ | $71.60 \pm 0.11$ |

concentrate around the decision boundary. The re-weighting dilutes the example frequency to both sides, thus the classification results are less sensitive to perturbation change.

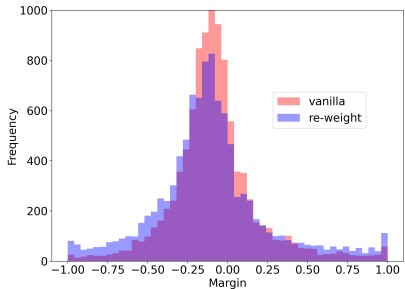

Figure 3: Distribution of margin: vanilla robust training vs re-weighting

### 4.2.2 EFFECTS OF AUTO-TUNING $\epsilon$

In robust training, a $\epsilon_{\text{train}}$ usually produces good defensive model for $\epsilon_{\text{eval}}$ which is slightly smaller than itself. Practitioners refrain from using an even larger $\epsilon$ due to the marginal gain of robustness accuracy with the cost of sacrificing natural accuracy.

The idea of using auto-tuning $\epsilon$ is to prevent improperly large perturbation for vulnerable points around decision boundary. When $\epsilon_{\text{train}}$ is considerably larger than the testing $\epsilon$, the worst case points exceed the decision boundary by a unrecoverable large margin due to model capacity. Our auto-tuning $\epsilon$ method trims the unnecessary perturbation to alleviate the counter effect for robust training.

Table 3: Robustness against different hyper-parameters with auto-tuning $\epsilon$ with IBP

| Dataset | $\epsilon_{\text{eval}}$ | $\epsilon_{\text{train}}$ | $\epsilon_{\text{maxoff}}$ | Clean err.(%) | Verified err.(%) |
|---|---|---|---|---|---|
| MNIST | 0.3 | 0.4 | baseline | $2.28 \pm 0.12$ | $8.66 \pm 0.05$ |
| | | | 0.05 | $2.08 \pm 0.08$ | $8.52 \pm 0.17$ |
| | | | 0.1 | $2.04 \pm 0.05$ | $8.46 \pm 0.23$ |
| | | | 0.15 | $2.04 \pm 0.10$ | $\mathbf{8.36 \pm 0.12}$ |
| | | | 0.2 | $2.17 \pm 0.02$ | $8.78 \pm 0.06$ |
| CIFAR-10 | 8/255 | 8.8/255 | baseline | $49.44 \pm 0.49$ | $\mathbf{72.26 \pm 0.16}$ |
| | | | 0.005 | $49.17 \pm 0.29$ | $72.75 \pm 0.53$ |
| | | | 0.01 | $49.21 \pm 0.74$ | $73.23 \pm 0.50$ |
| | | | 0.015 | $49.21 \pm 0.19$ | $73.37 \pm 0.27$ |
| CIFAR-10 | 8/255 | 14/255 | baseline | $55.13 \pm 0.73$ | $70.50 \pm 0.62$ |
| | | | 0.005 | $55.78 \pm 0.30$ | $\mathbf{69.29 \pm 0.58}$ |
| | | | 0.01 | $55.53 \pm 0.88$ | $69.44 \pm 0.43$ |
| | | | 0.015 | $55.62 \pm 0.34$ | $69.93 \pm 0.64$ |

Table 3 shows the hyper-parameters choice for auto-tuning $\epsilon$. For the MNIST data set, we found the optimal hyper-parameter $\epsilon_{\text{maxoff}}$ around 0.15 where the certified accuracy gains by 0.3% from baseline.

For the CIFAR data set, the standard $\epsilon_{\text{train}}$ 8.8/255 is only 10% larger than the $\epsilon_{\text{eval}}$. In this case, our auto-tuning $\epsilon$ method performs worse compared to the baseline results because the extra 10% is necessary for certified training. By using larger $\epsilon_{\text{train}}$, it is possible to reach even smaller testing error. To show the effect of auto-tuning $\epsilon$ method, we will increase the $\epsilon_{\text{train}}$ until the testing verified accuracy stabilizes. For $\epsilon_{\text{train}} = 10/255, 14/255, 18/255, 22/255$, we come up with verified accuracy $= 70.62\% \pm 0.62\%, 70.50\% \pm 0.62\%, 70.31\% \pm 0.83\%, 71.19\% \pm 0.21\%$ which are roughly 2% lower than the standard result. We choose $\epsilon = 14/255$ as the "sweet point", beyond which any increase in $\epsilon$ does not provide any improvement in accuracy because the very large perturbation is harmful to training. With the optimal $\epsilon_{\text{maxoff}} = 0.01$, the verified accuracy is improved by 1.21% from baseline.

### 4.2.3 COUPLING OF TWO METHODS

By coupling two methods, we can further combine the improvements and achieve even better results. As shown in Table 6, for MNIST, using both optimal hyper-parameters from above individual tests we reach a verified accuracy of $8.01\%$, gaining $0.65\%$ from baseline. For CIFAR-10, our best results beats the IBP baseline by $2.17\%$. We performed more experiments on other $\epsilon_{eval}$ and other architecture and defer the results to Appendix C.

Table 4: Robustness improvement summary with IBP

| *Dataset* | $\epsilon_{\text{eval}}$ | $\epsilon_{\text{train}}$ | re-weight | auto-eps | Clean err.(%) | Verified err.(%) |
|---|---|---|---|---|---|---|
| *MNIST* | 0.3 | 0.4 | baseline | | $2.28 \pm 0.12$ | $8.66 \pm 0.05$ |
| | | | | ✓ | $2.04 \pm 0.10$ | $8.36 \pm 0.12$ |
| | | | ✓ | | $2.30 \pm 0.07$ | $8.42 \pm 0.07$ |
| | | | ✓ | ✓ | $2.09 \pm 0.06$ | $\mathbf{8.01 \pm 0.04}$ |
| *CIFAR-10* | 8/255 | 14/255 | baseline | | $55.13 \pm 0.73$ | $70.50 \pm 0.62$ |
| | | | | ✓ | $55.66 \pm 1.05$ | $69.8 \pm 0.39$ |
| | | | ✓ | | $55.78 \pm 0.30$ | $69.29 \pm 0.58$ |
| | | | ✓ | ✓ | $55.05 \pm 0.17$ | $\mathbf{68.33 \pm 0.21}$ |

### 4.2.4 RESULTS ON CROWN-IBP

To illustrate that our methods generally work for all the certifiable training, we apply the same methods and optimal hyper-parameters from IBP setup directly to CROWN-IBP. For MNIST data, we achieves $0.32\%$ improvement from baseline by re-weighting and $0.13\%$ from auto-eps, combining the two is similar to using auto-eps only though. For CIFAR data, note that we use training batch size 256 instead of 1024 in Zhang et al. (2020a), the standard $\epsilon_{\text{train}}$ 8.8/255 has verified error $68.10\% \pm 0.12\%$ at $\epsilon_{\text{eval}} = 8/255$ from our experiments. By elevating the bulk $\epsilon_{\text{train}}$ to 14/255, the verified error increases to $68.35\% \pm 0.43\%$. After applying the two methods, we achieve $66.72\% \pm 0.7\%$ and beat the baseline by $1.38\%$.

Table 5: Robustness improvement summary with CROWN-IBP

| *Dataset* | $\epsilon_{\text{eval}}$ | $\epsilon_{\text{train}}$ | re-weight | auto-eps | Clean err.(%) | Verified err.(%) |
|---|---|---|---|---|---|---|
| *MNIST* | 0.3 | 0.4 | baseline | | $1.85 \pm 0.11$ | $7.02 \pm 0.08$ |
| | | | | ✓ | $1.62 \pm 0.01$ | $6.89 \pm 0.05$ |
| | | | ✓ | | $1.86 \pm 0.04$ | $\mathbf{6.70 \pm 0.30}$ |
| | | | ✓ | ✓ | $1.74 \pm 0.02$ | $6.90 \pm 0.05$ |
| *CIFAR-10* | 8/255 | 14/255 | baseline | | $54.29 \pm 0.73$ | $68.35 \pm 0.43$ |
| | | | | ✓ | $54.73 \pm 0.34$ | $67.12 \pm 0.08$ |
| | | | ✓ | | $53.41 \pm 0.42$ | $67.40 \pm 0.32$ |
| | | | ✓ | ✓ | $53.65 \pm 0.89$ | $\mathbf{66.72 \pm 0.7}$ |

## 5 CONCLUSIONS

This paper proposed new methods to refine the decision boundary to improve the certifiable robustness accuracy. We prove the necessity to assign more weights towards adversarial examples around decision boundary by parameterizing the true adversarial distribution. A future direction could be exploring the adversarial mechanism for better approximation of re-weight function and promoting the idea of re-weighting to other non-adversarial tasks. For auto-tuning, our initial interest was to prevent extremely large $\epsilon_{\text{train}}$ and a relatively large $\epsilon_{\text{train}}$ is necessary to show the effect. In practice, we can increase $\epsilon_{\text{train}}$ until accuracy stabilizes and achieve further optimal verified error. Our intuitive approach indeed finds a proper perturbation for each example. Given the fact that a larger $\epsilon_{\text{train}}$ achieves optimal certifiable accuracy still exists as an empirical finding (Gowal et al., 2019; Zhang et al., 2020a), a rigorous theoretical thinking is needed to justify this approach.

## ETHICS STATEMENT

This paper does not contain ethics concerns.

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

## APPENDIX

## A    PROOF OF THEOREM 1

**Theorem 1** (Restated). *Given a binary classification task $\mathbb{R}^n \rightarrow \{+, -\}$ with equal prior probability, assume the corresponding examples $\boldsymbol{x}_+$ and $\boldsymbol{x}_-$ are uniformly distributed in region $\mathbb{S}_+, \mathbb{S}_- \subset \mathbb{R}^n, \mathbb{S}_+ \cap \mathbb{S}_- = \emptyset$. If there exists a bijection mapping $m := \mathbb{S}_+ \mapsto \mathbb{S}_-, \mathbb{S}_- \mapsto \mathbb{S}_+$ s.t. the post-mapping example $\boldsymbol{x}'_+$ and $\boldsymbol{x}'_-$ distribution remains uniform in each region, the expectation of conditional distribution ratio $\frac{P'(\boldsymbol{x}'|y)}{P(\boldsymbol{x}|y)}$ over original distribution $P(\boldsymbol{x}, y)$ is great or equal than 1.*

*Proof.* let $\boldsymbol{x}_+, \boldsymbol{x}_-, \boldsymbol{x}'_+, \boldsymbol{x}'_-$ denote the pre-mapping and post-mapping examples for 2 labels, because both pre-mapping and post-mapping examples are uniformly distributed in each region, $P(\boldsymbol{x}_+|y_+), P(\boldsymbol{x}_-|y_-), P'(\boldsymbol{x}'_+|y_+), P'(\boldsymbol{x}'_-|y_-)$ are constant. let $S_+ = \oint_{\boldsymbol{x} \in \mathbb{S}_+} d\boldsymbol{x}, S_- = \oint_{\boldsymbol{x} \in \mathbb{S}_-} d\boldsymbol{x}$ be the "volume" of set $\mathbb{S}_+, \mathbb{S}_-$.
Given total conditional probability equals 1:

$$\int_{\boldsymbol{x} \in \mathbb{S}_+} P(\boldsymbol{x}_+|y_+) d(\boldsymbol{x}) = 1$$

$$\Longleftrightarrow P(\boldsymbol{x}_+|y_+) \int_{\boldsymbol{x} \in \mathbb{S}_+} d(\boldsymbol{x}) = 1$$

$$\Longleftrightarrow P(\boldsymbol{x}_+|y_+) = 1/S_+$$

Similarly we have other conditional probability equals the inverse of volume.
Hence,

$$\mathbb{E}_{(\boldsymbol{x},y) \sim P}\left[\frac{P'(\boldsymbol{x}'|y)}{P(\boldsymbol{x}|y)}\right] = P(y_+) \int_{\boldsymbol{x} \in \mathbb{S}_+} P(\boldsymbol{x}_+|y_+) \frac{P'(\boldsymbol{x}'_+|y_+)}{P(\boldsymbol{x}_+|y_+)} d(\boldsymbol{x})$$

$$+ P(y_-) \int_{\boldsymbol{x} \in \mathbb{S}_-} P(\boldsymbol{x}_-|y_-) \frac{P'(\boldsymbol{x}'_-|y_-)}{P(\boldsymbol{x}_-|y_-)} d(\boldsymbol{x})$$

$$= 0.5\left(\frac{S_+}{S_-} + \frac{S_-}{S_+}\right) \geq 1$$

where the second equality comes from the balanced prior assumption($P(y_+) = P(y_-)$)

$\square$

Notice we cannot directly cancel the conditional probability on the numerator and denominator, otherwise integrating $\boldsymbol{x}'$ over $d\boldsymbol{x}$ is not tractable. The inequality holds when $S_- \neq S_+$.

# B   A CONCRETE EXAMPLE ON IMPORTANCE WEIGHT EXPECTATION

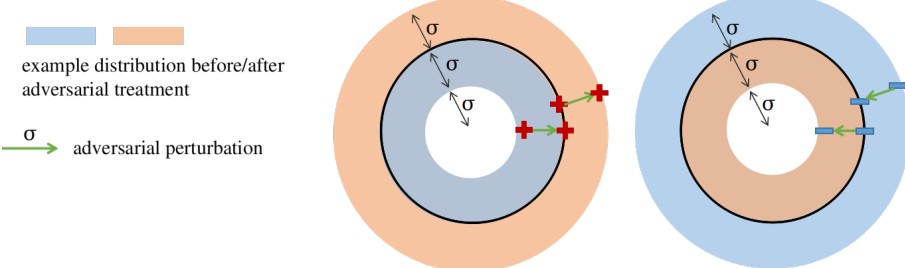

Figure 4: An ideal example illustrating importance weight expectation around decision boundary with curvature

**Example 2.** We assume label-balanced data $\boldsymbol{x} \in \mathbb{R}^2$ and $y \in \{+, -\}$ and positive constant $\sigma$, the clean data with label 1 were uniformly distributed in the inner ring $C_{in} = \{\boldsymbol{x}|\sigma < \|\boldsymbol{x}\|_2 < 2\sigma\}$ and data with label 0 were uniformly distributed in the outer ring $C_{out} = \{\boldsymbol{x}|2\sigma < \|\boldsymbol{x}\|_2 < 3\sigma\}$. Then $p(y = +) = p(y = -) = 0.5$ and clean data distribution is: $p(\boldsymbol{x}|y) = \begin{cases} \frac{1}{3\pi\sigma^2} & \text{if } y = +, \sigma < \|\boldsymbol{x}\|_2 < 2\sigma \\ \frac{1}{5\pi\sigma^2} & \text{if } y = -, 2\sigma < \|\boldsymbol{x}\|_2 < 3\sigma \\ 0 & \text{otherwise} \end{cases}$
assume the linear classifier $f : \mathbb{R} \to \mathbb{R}$ uses $r = \|\boldsymbol{x}\|_2$ as feature and outputs the logit for probability,

s.t. the probability after sigmoid function is $p_f(y|\boldsymbol{x}) = \begin{cases} \frac{1}{1+e^s} & \text{if } y = + \\ \frac{1}{1+e^{-s}} & \text{if } y = - \end{cases}$

where $s = r + 2\sigma$
The negative log-likelihood

$$NLL = -\mathbb{E}_{(\boldsymbol{x},y)\sim p(\boldsymbol{x},y)}[log(p_f(y|\boldsymbol{x}))]$$

$$= -p(y = +) \int p(\boldsymbol{x}|y = +)log(p_f(y = +|\boldsymbol{x}))d\boldsymbol{x}$$

$$- p(y = -) \int p(\boldsymbol{x}|y = -)log(p_f(y = -|\boldsymbol{x}))d\boldsymbol{x}$$

$$= -0.5(\frac{1}{3\pi\sigma^2}\int_{r=\sigma}^{r=2\sigma}\frac{1}{1+e^s}2\pi rdr + \frac{1}{5\pi\sigma^2}\int_{r=2\sigma}^{r=3\sigma}\frac{1}{1+e^{-s}}2\pi rdr)$$

$$= -0.5(\frac{1}{3\pi\sigma^2}\int_{s=-\sigma}^{s=0}\frac{1}{1+e^s}2\pi(s+2\sigma)ds + \frac{1}{5\pi\sigma^2}\int_{s=0}^{s=\sigma}\frac{1}{1+e^{-s}}2\pi(s+2\sigma)ds)$$

Notice the integral includes polylogarithm term thus the indefinite integral is not elementary function and it is unnecessary to attack the final integral. Instead, we apply adversarial directly to the above equation for $(\boldsymbol{x}, y)$ distribution. Calculate the adversarial gradient: for $s \in [0, \sigma]$, $\frac{dNLL(s)}{ds} = -\frac{1}{6\pi\sigma^2}\frac{e^s}{1+e^{-s}}2\pi(s+2\sigma)$, let $\Delta_-(s) = \frac{d^2NLL(s)}{ds^2} = -\frac{1}{3\sigma^2}\frac{1+e^{-s}+e^{-s}(s+2\sigma)}{(1+e^{-s})^2}$. For $s \in [-\sigma, 0]$, $\frac{dNLL(s)}{ds} = -\frac{1}{6\pi\sigma^2}\frac{e^{-s}}{1+e^{-s}}2\pi(s+2\sigma)$, let $\Delta_+(s) = \frac{d^2NLL(s)}{ds^2} = -\frac{1}{3\sigma^2}\frac{1+e^s-e^s(s+2\sigma)}{(1+e^s)^2}$.

$\Delta_+(s)$ and $\Delta_-(s)$ denotes the attack gradient towards negative log-likelihood function on example $(\boldsymbol{x}, y)$ with $s = \|\boldsymbol{x}\|_2 + 2\sigma$. Notice $\Delta_+(s) > 0$ and $\Delta_-(s) < 0$ because adversarial perturbs the correctly classified examples towards the decision boundary. Assume the attack step $\alpha = \sigma$, there exist a mapping of attack $m : \mathbb{R}^2 \times \mathbb{R} \to \mathbb{R}^2 \times \mathbb{R}$. Because this is a 1 step attack on a 2D plane with gradient towards radial direction without projection, the mapping also corresponds to a verifiable worst case adversarial with $\epsilon_{max} = \sigma$ under $\ell_2$ norm. Rewrite $(\boldsymbol{x}, y)$ in the polar coordinate$(r, \theta, y)$,
$(r', \theta', y') = m(r, \theta, y) = \begin{cases} (r + \sigma, \theta, y) & \text{if } y = + \\ (r - \sigma, \theta, y) & \text{if } y = - \end{cases}$

After the attack, the importance ratio $\frac{P'(\boldsymbol{x}'|y)}{P(\boldsymbol{x}|y)} = \frac{r}{r'}$ is inversely proportional to the radius change.

The expected importance ratio could be written as :

$$
\begin{aligned}
E_{(x,y)\sim P}[\frac{P'(\boldsymbol{x}'|y)}{P(\boldsymbol{x}|y)}] &= P(y=-) \int P(\boldsymbol{x}|y=-)\frac{P'(\boldsymbol{x}'|y=-)}{P(\boldsymbol{x}|y=-)}d\boldsymbol{x} \\
&\quad + P(y=+) \int P(\boldsymbol{x}|y=+)\frac{P'(\boldsymbol{x}'|y=+)}{P(\boldsymbol{x}|y=+)}d\boldsymbol{x} \\
&= 0.5[\int_{r=2\sigma}^{r=3\sigma} \frac{1}{5\pi\sigma^2}\frac{r}{r-\sigma}2\pi r dr + \int_{r=\sigma}^{r=2\sigma} \frac{1}{3\pi\sigma^2}\frac{r}{r+\sigma}2\pi r dr] \\
&= 0.5[\int_{t=2\sigma-\sigma}^{t=3\sigma-\sigma} \frac{1}{5\pi\sigma^2}\frac{\sigma+t}{t}2\pi(\sigma+t)dt \\
&\quad + \int_{t=\sigma+\sigma}^{t=2\sigma+\sigma} \frac{1}{3\pi\sigma^2}\frac{-\sigma+t}{t}2\pi(-\sigma+t)dt] \\
&= \pi[\frac{1}{5\pi\sigma^2}(\sigma^2 log(t)+2\sigma t + t^2/2)\Big|_{t=2\sigma-\sigma}^{t=3\sigma-\sigma} \\
&\quad + \frac{1}{3\pi\sigma^2}(\sigma^2 log(t)-2\sigma t + t^2/2)\Big|_{t=\sigma+\sigma}^{t=2\sigma+\sigma} \\
&= \frac{1}{5\sigma^2}(\sigma^2 log(2)+3.5\sigma^2) + \frac{1}{3\sigma^2}(\sigma^2 log(1.5)+0.5\sigma^2) \\
&\approx 1.14 > 1
\end{aligned}
$$

## C MORE EXPERIMENTS

Table 6: Robustness improvement summary with IBP

| Dataset | architecture | $\epsilon_{\text{eval}}$ | $\epsilon_{\text{train}}$ | re-weight | auto-eps | Clean err.(%) | Verified err.(%) |
|---------|--------------|------------|-------------|-----------|----------|---------------|------------------|
| MNIST | DM-small | 0.3 | 0.4 | baseline | | $3.27 \pm 0.24$ | $12.00 \pm 0.35$ |
| | | | 0.4 | | ✓ | $3.25 \pm 0.10$ | $11.73 \pm 0.57$ |
| | | | 0.4 | ✓ | | $2.75 \pm 0.11$ | $11.36 \pm 0.24$ |
| | | | 0.4 | ✓ | ✓ | $2.87 \pm 0.10$ | $\mathbf{11.09 \pm 0.33}$ |
| CIFAR-10 | DM-small | 8/255 | 8.8/255 | baseline | | $51.86 \pm 0.18$ | $73.92 \pm 0.17$ |
| | | | 14/255 | baseline | | $58.08 \pm 0.33$ | $71.07 \pm 0.23$ |
| | | | 14/255 | | ✓ | $57.56 \pm 0.76$ | $70.33 \pm 0.29$ |
| | | | 14/255 | ✓ | | $58.36 \pm 1.05$ | $70.62 \pm 0.46$ |
| | | | 14/255 | ✓ | ✓ | $57.29 \pm 0.17$ | $\mathbf{70.02 \pm 0.18}$ |
| MNIST | DM-large | 0.1 | 0.2 | baseline | | $1.06 \pm 0.07$ | $3.08 \pm 0.15$ |
| | | | 0.2 | | ✓ | $1.03 \pm 0.01$ | $3.06 \pm 0.02$ |
| | | | 0.2 | ✓ | | $1.14 \pm 0.07$ | $2.86 \pm 0.09$ |
| | | | 0.2 | ✓ | ✓ | $1.03 \pm 0.04$ | $\mathbf{2.92 \pm 0.07}$ |
| CIFAR-10 | DM-large | 2/255 | 2.2/255 | baseline | | $33.89 \pm 0.42$ | $58.80 \pm 0.84$ |
| | | | 6/255 | baseline | | $44.32 \pm 0.27$ | $52.93 \pm 0.42$ |
| | | | 6/255 | | ✓ | $44.01 + \pm0.46$ | $52.38 \pm 0.10$ |
| | | | 6/255 | ✓ | | $43.60 \pm 0.18$ | $52.57 \pm 0.30$ |
| | | | 6/255 | ✓ | ✓ | $42.94 \pm 0.64$ | $\mathbf{52.17 \pm 0.17}$ |

