# OpenReview forum: "Tactics on Refining Decision Boundary for Improving Certification-based Robust Training"
_ICLR.cc/2022/Conference — ICLR 2022 Submitted_

### Official Review · Reviewer_jhkj · 2021-10-18

**Correctness:** 3
**Technical Novelty And Significance:** 3
**Empirical Novelty And Significance:** 3
**Recommendation:** 6
**Confidence:** 4

**Main Review:**

The paper mainly proposes two tactics: training data reweighting, and sample-level $\varepsilon$ tuning. The training data reweighting is backed up by theoretical justifications and intuitive explanations. The sample-level $\varepsilon$ tuning is relatively intuitive. Both tactics contribute to a better certified robust training approach.

Strengths:
- The proposed two tactics are novel and solid.
- The theoretical and empirical illustrations of the proposed method are insightful.
- Demonstrated by experimental results, the proposed method is effective. The relaxation-based training is hard to improve where recent work like CROWN-IBP, Auto-LiRPA, etc can only get 1%-3% improvements despite their complex techniques. This work is simple but has shown a similar level of improvement.

Weaknesses:
- The experimental evaluation is not comprehensive enough. The following results would be helpful if provided:

  - Experiments on more model architectures other than DM-Large to show generalizability.

  - Combine the proposed method with Auto-LiRPA [1], since Auto-LiRPA achieves a lower certified error compared with IBP and CROWN-IBP, e.g., 66.62% on CIFAR-10 against 8/255 $\ell_\infty$ perturbations.

  - Show running time statistics to quantitatively justify the small overhead of the proposed method.

  - Provide code for replicating the experiments, or provide a reproducibility statement.

  - How many times did you replicate the experiments? It would be good to report this number in the paper.

- Several writing issues as listed below.

Minor:
1. The paper title uses "certification-based" while the first sentence in the abstract uses "verification-based". Please make them consistent.
2. Section 1, first paragraph: FGSM were the very first approaches => was the very first approach.
3. Section 3.1: ${\mathbb{E}}\_{(\mathbf{x}',y) \sim P} [l(f_{\theta}(\mathbf{x}'), y)]$ => $\mathbb{E}\_{(\mathbf{x}',y)\sim P'} [l(f_{\theta}(\mathbf{x}'), y)]$
4. Example1: and vice versa for examples with examples with ′+′ label moving to the external arc => and vice versa for examples with ′+′ label moving to the external arc
5. Algorithm 1 in Appendix C: How does Algorithm 1 combine with the whole training pipeline? Is there any training process that optimizes the model weights at the end of each inside loop?

Suggestions:
1. It would be good to discuss randomized smoothing and related training (Cohen et al ICML 2019, Lecuyer et al S&P 2019, Salman et al NeurIPS 2019), Lipschitz-based robustness and related training (Tsuzuku et al NIPS 2018, Trockman et al ICML 2021) in related work. They also provide verifiable robustness.
2. It would be good to move the algorithm description in Appendix C to the main text, otherwise the hyperparameter $\varepsilon_{maxoff}$ shown in Section 4.2.2 has no formal definition.

Question:
When computing the sample weights, what $\varepsilon$ is used for IBP to compute the margin (Eq. 11)? Is it the original $\varepsilon$ or the auto-tuned $\varepsilon$? If it is former, I worry the IBP bound is too loose for hard samples with large original $\varepsilon$, where maybe smaller $\varepsilon$ could be better. If it is latter, the auto-tuned $\varepsilon$ may deviate too much from the original required $\varepsilon$, failing to precisely reflect the actual margin bound under the original required $\varepsilon$.

[1] Xu, Kaidi, et al. "Automatic perturbation analysis for scalable certified robustness and beyond." Advances in Neural Information Processing Systems 33 (2020).

**Summary Of The Paper:**

This paper proposes to integrate training data reweighting and sample-level perturbation budget tuning to improve relaxation-based robust training like IBP and CROWN-IBP. The overhead is small and the improvement of the certified accuracy is +1.38% on CROWN-IBP and +2.17% on IBP on CIFAR-10.

**Summary Of The Review:**

The proposed two tactics are backed up by intuitive but useful insights and shown effective for relaxation-based training. Therefore, the work would be inspirational to the community.
However, the current experimental evaluation is not comprehensive enough and lacks several important details, so it is a bit difficult to tell whether the approach is generally effective or not. Also, the presentation can be improved.
Given these concerns, I think the paper is marginally below the threshold. If more experimental details are provided during the discussion phase, I am happy to re-evaluate this paper.

---

> ### Author Response · Authors · 2021-11-19
> **Author response 1/2**
>
> Thank you so much for the detailed reviews and constructive suggestions. We have $\textbf{conducted the suggested experiments, provided the source code and showed the running statistics}$. Our response is listed as following:
>
> $\textbf{1. Experiments}$:
>
> $\bullet$ Other architectures:
>
> We have performed more experiments on DM-small following your suggestion.
> $\text{Table R3(restated): DM-large, MNIST, IBP, $\epsilon_{test}=0.1$}$
> $$
> \begin{array} {|r|r|r|r|r|}
> \hline \epsilon_{train}  & \text{re-weight} & \text{auto-eps}& \text{clean err.($\\%$) } & \text{verified err.($\\%$)  }  \\\\
> \hline 0.4 &  & & 3.27\pm0.24 &  12.00\pm0.35 \\\\
> \hline 0.4 &  \checkmark & & 3.25\pm0.10 &  11.73\pm0.57 \\\\
> \hline 0.4 &  & \checkmark & 2.75\pm0.11 &  11.36\pm0.24 \\\\
> \hline 0.4 &  \checkmark & \checkmark  & 2.87\pm0.1 &  11.09\pm0.33 \\\\
> \hline  \end{array}
> $$
>
> $\text{Table R4(restated): DM-small, CIFAR-10, IBP, $\epsilon_{test}=8/255$}$
>
> $$
> \begin{array} {|r|r|r|r|r|}
> \hline \epsilon_{train}  & \text{re-weight} & \text{auto-eps}& \text{clean err.($\\%$) } & \text{verified err.($\\%$)  }  \\\\
> \hline 8.8/255 &   & & 51.86 \pm 0.18 &  73.92\pm 0.17 \\\\
> \hline 14/255 &  & & 58.08\pm0.33 &  71.07\pm0.23 \\\\
> \hline 14/255 &  \checkmark & & 57.56\pm0.76 &  70.33\pm0.29 \\\\
> \hline 14/255 &  & \checkmark & 58.36\pm1.05 &  70.62\pm0.46 \\\\
> \hline 14/255 &  \checkmark & \checkmark  & 57.29\pm0.17 &  70.02\pm0.18 \\\\
> \hline  \end{array}
> $$
>
>
> We have also performed experiments on other $\epsilon_{test}$ to show the generalizability of our approaches. Please refer to our reply for Reviewer $\textbf{onQv}$ for detailed results. The performance/improvement is consistent with our results in the main paper.
>
> $\bullet$  Auto-LiRPA :
> Thank you for the suggestion on Auto-LiRPA. We note that our method is compatible to the AutoLiRPA. Auto-LiRPA is a computational package extended from relaxation perturbation methods such as CROWN-IBP. Its novelty lies in the efficiency compared with CROWN-IBP, the forward/backward algorithm is similar. In addition, we found that it’s computation heavy to run with Auto-LiRPA, so we use IBP/CROWN-IBP as the major comparison baselines.
>
> $\bullet$  Running time statistics:
> We tested 4 experiments with CIFAR task of 3200 epochs on a single Nvidia V100 graphic card with reweight/auto-eps feature enabled/disabled.  Each experiment is performed 3 times to get the mean/variance of running time. Re-weighting and auto-eps costs additional ~0.6min and 6 min respectively, but this is small compared with the total running time ~340min. Auto-eps consumes more time because it requires calling memory to record the customized eps for each individual example.
>
> $\text{Table R6:Running time statistics}$
> $$
> \begin{array} {|r|r|r|r|r|}
> \hline  \text{re-weight} & \text{auto-eps}& \text{running time(min) }   \\\\
> \hline  & & 335.7\pm1.6 \\\\
> \hline   \checkmark & & 336.3\pm1.0 \\\\
> \hline  & \checkmark & 341.8\pm0.4  \\\\
> \hline  \checkmark & \checkmark  & 342.3\pm0.4 \\\\
> \hline  \end{array}
> $$
>
> $\bullet$  code/reproducibility:
>
> In the main paper we reported all the experiments are conducted 3 times in different seeds. We have also attached our source code in the supplementary material to replicate the results. We anonymized the source code except some Copyrights of the original author of CROWN-IBP codes. Please bear with us the code is not cleaned up due to limited rebuttal time, all the experiment scripts are organized in “train_all_configs.sh”.
>
> $\textbf{2. writing issues}$:
>
> $\bullet$ Typos: Thank you for the detailed comment, we followed most of your suggestions and highlighted it in the manuscript.
>
> $\bullet$ Algorithm 1:
> Algorithm 1 is to illustrate the procedure to update customized eps for each example. In the training pipeline, we use the current eps to calculate the $Z_K$ and loss function and perform optimization step on model parameters. Then we use the $Z_K$  to update eps for next step use. We have also moved Algorithm 1 to the main paper following your suggestion.
>
> $\textbf{3. Other certification methods}$:
>
> Thank you for the constructive suggestion, we included your suggested discussions in the new manuscript.

---

> > ### Comment · Reviewer_jhkj · 2021-11-20
> > **Follow-up Discussion**
> >
> > Thanks to the authors for the detailed reply and conducted additional experiments.
> > Here are some follow-up feedbacks, questions, and concerns.
> >
> > 1. experiments:
> > I am satisfied with the provided results given the tight rebuttal time window.
> > To provide a more comprehensive study, in the next version of the paper, I would suggest the authors combine the proposed tactics with Auto-LiRPA (leverage the loss fusion mode), L-$\infty$ net [1], and Cayley orthogonal networks [2], and further evaluate on TinyImageNet to show the scalability since the proposed tactics are quite general. (note: adding them or not at this stage will not affect my score)
> >
> >
> > 2. writing issues: there are still some typos in the revised version. For example:
> > (1) Page 1, added related work: random smoothing -> random**ized** smoothing. **where** in this paper we mainly focus on -> **on which** in this paper we mainly focus
> > (2) Page 6, end of Section 3.2: The algorithm is shown in **Algorithm 3.2** -> **Algorithm 1**
> >
> > Algorithm 1 is still unclear to me because it has an outer loop that iterates over epochs. Based on my understanding, inside each epoch and inside each training batch loop, the algorithm should first use $\varepsilon_i$ to **train** the model, then conduct the following steps. Otherwise, the algorithm is not formally correct as a continuous execution procedure, since the model $f_{\theta}$'s parameters $\theta$ is never changed since its initialization.
> >
> > 3. Other certification methods: I appreciate the added discussion though it contains a few typos as mentioned above.
> >
> > 4. $\varepsilon$ used for margin calculation: it makes sense to use original/nominal $\varepsilon$. Then, my follow-up question is, starting from which epoch does the auto-tuning come into play? My intuition is that, if we auto-tune the $\varepsilon$ from the beginning or from too-early epochs, whether the current training sample would lie in the inferior or close to the boundary may be non-stable or non-indicative since the warm-up $\varepsilon$ is too slow and the model is not well-trained yet.
> >
> >
> > [1] Zhang, Bohang, et al. "Towards Certifying L-infinity Robustness using Neural Networks with L-inf-dist Neurons." International Conference on Machine Learning. PMLR, 2021.
> > [2] Trockman, Asher, and J. Zico Kolter. "Orthogonalizing convolutional layers with the cayley transform." ICLR 2021.

---

> > > ### Author Response · Authors · 2021-11-21
> > > **Author response to follow-up discussion**
> > >
> > > Thank you for the new comments and we really appreciate your quick feedback. We are also glad that you are pleased with the updated experiments. Following your addtional questions:
> > >
> > > * Experiments: thank you for your suggestions and we will use them as guidelines for the next version of this paper, we are expecting a similar trend on larger scales.
> > > * Algorithm 1: we recognized it is indeed necessary to include the training step, now we have updated the algorithm with optimization steps in the new manuscript.
> > > * Typos: Thank you again for pointing out the typos, we corrected it.
> > > * Auto-eps warm-up time: the auto-eps algorithm has the same warm-up time with nominal $\epsilon$ schedule, when nominal $\epsilon$ starts growing the auto-eps kicks in. We agree with your claim that at the very beginning of the $\epsilon$ schedule, the adversarial margin is close to the clean example margin and its sign is not indicative of robustness. While in this stage, the $\epsilon$ is small and lower bounded by 0 and the maximum variation of $\epsilon_{train}$ is limited by the absolute value of $\epsilon$. Besides, during the warm up stage, the model is already well-trained for clean data. At the beginning of the standard $\epsilon$ schedule, a significant portion of examples are certified robust and a small portion of examples requires non-zero $\epsilon_{off}$. Therefore, a slight change of $\epsilon_{off}$ has limited effect on training. In this case, the training process is not prone to the instability caused by auto-eps at this early stage.

---

> > > > ### Comment · Reviewer_jhkj · 2021-11-21
> > > > **Thanks for your response**
> > > >
> > > > Thanks for your response. Most of my concerns are resolved.
> > > >
> > > > But let me double-check whether my understanding about $\epsilon$ tuning is correct: when you say "we use the original/nominal $\varepsilon$ in the schedule", do you mean using $\varepsilon_{base}$ defined in Algorithm 1 to compute the margin? It seems to be contradicting the pseudo-code which uses $\varepsilon_i = \varepsilon_{base} - \varepsilon_{i,off}$.

---

> > > > > ### Author Response · Authors · 2021-11-21
> > > > > **Author response, corrected reply**
> > > > >
> > > > > Thank you again for the quick feedback. We double-checked our code and found our original reply was incorrect. Actually, the auto-eps tuned $\epsilon$ instead of the nominal $\epsilon$ is used for margin calculation. $\textbf{The algorithm in the draft is consistent with our implementation but not our first rebuttal reply}$. We are really sorry for the confusion in the original response!
> > > > >
> > > > > We would like to correct the answers as follows:
> > > > >
> > > > > Regarding the legitimacy of using different $\epsilon$ as margin metric, we admit this is not fair among different examples. While during the training process, the $\epsilon_{off}$ is strongly correlated with margin($\epsilon_{off}$ of current epoch is proportional to the absolute value of margin in last epoch for uncertified examples). If a hard example is treated with large $\epsilon_{off}$ and smaller $\epsilon=\epsilon_{base}-\epsilon_{off}$ and its margin becomes small, $\epsilon_{off}$ will be auto-tuned to smaller value after this epoch. Therefore, when the training stabilizes, harder examples tend to have a larger negative margin(close to -1) and larger $\epsilon_{off}$ than easier examples. The margin metric under auto-tuned $\epsilon$ is monotonic with the “difficulty” of examples, while this relationship is not that straight-forward like using the nominal $\epsilon_{base}$.
> > > > >
> > > > > We also understand your concern that auto-tuned $\epsilon$ does not reflect the decision confidence under nominal $\epsilon_{base}$ from the scheduler when $\epsilon_{off}$ is large. Indeed, we are aiming at the model performance under $\epsilon_{test}$ instead of $\epsilon_{base}$ or $\epsilon_{train}$. In our setup, there exists an upper bound $\epsilon_{max, off}$ for $\epsilon_{off}$ and $\epsilon_{train}$ is greater than $\epsilon_{test}$. The actual $\epsilon$ used for margin calculation will not be far away from $\epsilon_{test}$. Combining this and our explanation in the above paragraph, we claim auto-tuned $\epsilon$ still reflects the “difficulty” of examples and could be used in margin calculation for auto-eps or re-weighting.
> > > > >
> > > > > We are again sincerely sorry for our incorrect reply during the rebuttal stage. But we are also glad to hear our previous replies addressed most of your concerns. We hope our explanation could clear up your concern on the $\epsilon$ used for margin calculation as well. Please do not hesitate to contact us if you need further clarification. We hope to address all your concerns and you will consider re-evaluating our work.

---

> > > > > > ### Comment · Reviewer_jhkj · 2021-11-22
> > > > > > **Follow-up reply**
> > > > > >
> > > > > > Thanks for your prompt reply.
> > > > > >
> > > > > > The interactions between re-weighting and auto-eps make it a bit hard to get an intuition on the training dynamics, and I can see the approach tries to eliminate possible negative correlation between these two tactics by limiting $\varepsilon_{max,off}$ and choosing $\varepsilon_{train} \gg \varepsilon_{test}$, and at least for hard samples when the training stabilizes, the tendency of weights and actual training $\varepsilon_i$ seem to be correct.
> > > > > >
> > > > > > Therefore, my concerns are resolved and I have increased the score by 1.

---

> > > > > > > ### Author Response · Authors · 2021-11-22
> > > > > > > **Thank you for increaing the score**
> > > > > > >
> > > > > > > Thank you again for your dedication in reviewing our paper and updating the score. We really appreciate your comments and suggestions and enjoy the discussion with you.
> > > > > > >
> > > > > > > All the best, Authors

---

> ### Author Response · Authors · 2021-11-19
> **Author response 2/2**
>
> $\textbf{4.  $\epsilon$ used for margin calculation}$:
>
> Thank you for the concern on which $\epsilon$ should be used to calculate margin. In practice we use the original/nominal $\epsilon$ in the scheduler. Auto-tuned $\epsilon$ is not legit because the margin metric is not fair among the examples. For instance, more vulnerable examples may be measured by a smaller $\epsilon$ so their confidence margin could be even greater than robust examples with larger $\epsilon$. But we understand your concern on using the original margin, it could be too large for very vulnerable examples. In this case, the absolute value of margin will be limited by 1 and they will be given a small weight factor and assumed “far away from decision boundary”.
>
> $\textbf{5. summary}$
>
> We briefly summarized our above response to your comments below:
> * #1: Experiments: we conducted experiments as you suggested and clarified the necessity to apply our methods on Auto-LiRPA. We’ve also shown the computation time statistics and addressed your concern on reproducibility and provided the source code.
> * #2: writing issues: We corrected the typos and moved Algorithm 1 to the main paper as suggested.
> * #3: We included more discussions on related papers.
> * #4: We discussed the $\epsilon$ used for margin calculation.

---

### Official Review · Reviewer_2goi · 2021-10-28

**Correctness:** 3
**Technical Novelty And Significance:** 2
**Empirical Novelty And Significance:** 2
**Recommendation:** 5
**Confidence:** 5

**Main Review:**

**Strength**

The motivation of the algorithm is clear and easy to follow. The algorithm is general and easy to implement.

**Weakness**

1. [Theoretical Strength]
    * From my point of view, the bijection assumption in Theorem 1 is too strong and is disconnected with the practices. This is because the adversarial perturbations are defined as the "imperceptible" perturbations, so the perturbed input of one class should not cover the support of another class. The perturbation defined as the bijection mapping here changes the semantic meanings of the input for sure and should not be considered "imperceptible".
    * In the first paragraph of Section 3.1, the distribution P' should depends on the model parameters, because adversarial examples are generated specifically for a model. To be rigorous, the authors should point this out.

2. [Algorithm Design]
    * The sentence below Equation (12) is incorrect, because the margin itself is a bound. The margin defined in (12) is negative does not necessarily mean the input is adversarial vulnerable, it only means the current input cannot be certified robust.
    * The exponential of the margin does not imply the right hand side in formula (13). $s$ is defined as the ratio of the adversarial and clean conditional probability, which shows no connection to the IBP margin. In addition, the tightness of IBP margin differs from instance to instance.
    * The algorithm has too many additional hyper-parameters, including $\alpha$, $\gamma$ and $\epsilon_{I, maxoff}$. The authors should conduct ablation study to show how sensitive the performance is to these additional hyper-parameters and whether or not the optimal hyper-parameters are too sensitive to the task and model architectures. I believe this will greatly facilitate the practitioners.

3. [Experiments]
    * The strength of the baselines are questionable and the results of the baselines are not consistent with the previous works. For example, the Table 3 in the IBP paper (Ref[A]) shows the IBP certified error on MNIST ($\epsilon = 0.3$) is $8.21%$ and on CIFAR10 ($\epsilon = 8 / 255$) is $68.44%$; both are better than the baseline results shown in Table 2 and Table 3 of this paper. The CROWN-IBP results in the original paper (Ref[B]) on CIFAR10 ($\epsilon = 8 / 255$) is $66.94$ in their Table 3 and is better than the reported results in Table 5 of this paper. I agree that the training $\epsilon$ here is a bit different, but the authors should present the baseline results in their optimal hyper-parameters settings.

4. [Writing and Presentation]
    * Some claims in this paper is not correct. For example, in the last paragraph of page 1, the authors say the incomplete certifier provides looser bounds by convex adversarial polytope. This is not comprehensive, because some other incomplete certifier uses semidefinite programming (Ref[C]) or randomized smoothing (Ref[D]). Convex polytope is only one category of the incomplete certifiers.
    * The definition of formula (4) is wrong. We should use the lower bound for the true labels and the upper bounds for the rest.

Reference

Ref[A]: Gowal, S., Dvijotham, K., Stanforth, R., Bunel, R., Qin, C., Uesato, J., ... & Kohli, P. (2018). On the effectiveness of interval bound propagation for training verifiably robust models. arXiv preprint arXiv:1810.12715.

Ref[B]: Zhang, H., Chen, H., Xiao, C., Gowal, S., Stanforth, R., Li, B., ... & Hsieh, C. J. (2019, September). Towards Stable and Efficient Training of Verifiably Robust Neural Networks. In International Conference on Learning Representations.

Ref[C]: Raghunathan, A., Steinhardt, J., & Liang, P. (2018, February). Certified Defenses against Adversarial Examples. In International Conference on Learning Representations.

Ref[D]: Cohen, J., Rosenfeld, E., & Kolter, Z. (2019, May). Certified adversarial robustness via randomized smoothing. In International Conference on Machine Learning (pp. 1310-1320). PMLR.

**Summary Of The Paper:**

This paper proposes a geometry-aware reweighting scheme and auto adjusted training adversarial budget to improve the performance of provably adversarial training. The intuition arises from the distance of the training examples to the decision boundary of the model.

**Summary Of The Review:**

Due to the concerns pointed above, I do not think the current manuscript is suitable for publication. However, I welcome the authors to clear my concerns in the rebuttal period. I will do a re-evaluation then.

---

> ### Author Response · Authors · 2021-11-19
> **Author response 1/2**
>
> Thank you so much for the valuable reviews, rigorous comments and constructive suggestions. We $\textbf{explained our theoretical approach and clarified the experiment baselines}$. We address your comments and concerns below.
>
> $\textbf{1. Theory}$
>
> $\bullet$ Bijection assumption:
>
> The assumption is imposed in order to simplify the problem and provide some theoretical grounds to illustrate the need of weighted loss. The purpose of stating this theorem is to present a set of extreme adversarial cases under basic two-class classification tasks. In a real world scenario, the perturbed examples from one class may never cover the other, but it is possible that a part of the examples within each class may swap their support zones. Therefore, the real world perturbation could be treated as a mixture of the extreme cases in this theorem and other mild cases where the perturbed examples did not cross the boundary. Besides, the goal of adversarial perturbation is to have the examples cross the decision boundary given certain neural network parameters, therefore it is not negligible to the neural network.
>
> $\bullet$ Distribution $P'$:
>
> Yes we agree with your rigorous claim. $P'$ is dependent on the bound method as well as the neural network parameters. We will denote it in the revision.
>
> $\textbf{2. Algorithm}$
>
> $\bullet$ Definition of margin:
>
> Thank you for pointing it out. We modified the sentence into “The margin is positive when the example can be certified robust, otherwise the margin is negative” and highlighted the changes(p.5).
>
> $\bullet$  Equation 13(Equation 7 in the revision pdf):
>
> The goal of parameterizing the weighting function is to approximate the expected conditional probability ratio between adversarial and clean example. We agree with your claim that this ratio differs from instance to instance, while it is difficult to calculate the exact ratio for each example. Therefore, we target the expected ratio over a distribution of examples with similar distance to the decision boundary instead of individual data points.  From our analysis of two examples and theorem 1, we found that when the examples are close to the boundary, the expectation of such a ratio is greater than the examples far away from the decision boundary. Therefore, we’d like to give more weight to those close to the decision boundary. And in the implementation, we use the absolute value of  “margin” as a measure of distance between perturbed examples and decision boundaries. To be rigorous, the right hand side should be the expectation of probability ratio over the examples with such a margin.
>
> $\bullet$ Hyper parameters:
>
> Yes we agree hyper-parameters are important for result improvements over the baseline but we found it is relatively easy to tune. We’ve shown our hyper-parameter study in table 2 and table 3. Although different $\alpha$ and $\gamma$ are used during the hyper-parameter tuning, we found the same optimal re-weighting hyperparameter across all the tasks. For the auto-eps tuning, the $\epsilon_{off,max}$ requires more empirical adjustment(the actual training eps cannot be too small, therefore $\epsilon_{off,max}$ must not be too large). In table 3, we found when $\epsilon_{train}$ is comparably larger than $\epsilon_{test}$, for instance MNIST with $\epsilon_{train}=0.4, \epsilon_{test}=0.3$, CIFAR-10 with $\epsilon_{train}=14/255, \epsilon_{test}=8/255$, verified err will first drop with $\epsilon_{off,max}$ growing from 0 and later when $\epsilon_{off,max}$ is too large it will rise again.

---

> ### Author Response · Authors · 2021-11-19
> **Author response 2/2**
>
> $\textbf{3. Experiment baseline}$:
> Thank you for pointing out the baseline issues, we produce the baseline results with the code provided by Ref[B].
>
> For MNIST IBP certified error with $\epsilon_{test}=0.3$, we found 8.21% in Ref[A] and 8.47% in Ref[B]. In our experiments we got 8.66%. For CIFAR-10 with $\epsilon_{test}=8/255$, as mentioned in Ref[B] table 2, Ref[A]’s 68.44% requires adding an extra PGD adversarial training loss. Without PGD, Ref[A] achieves 73.52%. In Ref[B], it used 3 different $\kappa$ schedules and achieves optimal(70.81%) when $\kappa $(1->0.0) for this specific task(CIFAR-10,DM-large,eps=8/255). While the optimal $\kappa$ differs among different tasks in Ref[B], $\kappa$(1->0.5) always achieves optimal clean accuracy. Therefore, we choose use a consistent $\kappa$ schedule(1->0.5) among all our experiments. Under this schedule, Ref[B] achieves 72.68% and we get 72.26% as baseline. Our IBP baselines are consistent with Ref[B].
>
> Regarding the CROWN-IBP baseline, we explained the setup in the paper. Due to memory limitations of our hardware(Nvidia V100 with 32GB memory), we use a batch size of 256 instead of 1024 in Ref[B](TPUv2 32 cores with 256GB memory). The other differences come from the $\kappa$ schedule, the optimal in Ref[B] was achieved when $\kappa$ mains 0, but we use a consistent $\kappa$ schedule(1->0.5) among all our experiments. Due to these two reasons, our baseline(68.10%)is lower than their optimal(66.94%) with $\kappa$ schedule(0->0.), but better than their $\kappa$ schedule(1->0.5)(69.55%). To show the effect of auto-eps tuning, we elevated the $\epsilon_{train}$ to 14/255, therefore we have a (68.35%) as baseline for $\epsilon_{train}=14/255$.
>
> We have summarized the baseline comparison for verified accuracy($\\%$)  in the following table:
>
> $\text{Table R5: experiment baseline comparison}$
> $$
> \begin{array} {|r|r|r|r|}
> \hline \text{Experiments }   & \text{Ref[A] }  & \text{Ref[B] } & \text{Ours }   \\\\
> \hline \text{IBP MNIST }  \epsilon_{test}=0.3& 8.21   & \kappa : 1\rightarrow 0.5: 8.47&  \kappa : 1\rightarrow 0.5: 8.66   \\\\
>  &  & \kappa : 0\rightarrow 0:9.76 &  &   \\\\
>  &  & \kappa : 1\rightarrow 0:8.73 &  &   \\\\
> \hline \text{IBP CIFAR-10 }  \epsilon_{test}=8/255 & \text{w/ PGD loss}: 68.44 & \kappa : 1\rightarrow 0.5: 72.68 & \kappa : 1\rightarrow 0.5: 72.26 \\\\
>  &    \text{w/o PGD loss}: 73.52 & \kappa : 1\rightarrow 0: 70.81 &  \\\\
>  &  &    \kappa : 0\rightarrow 0: 71.22 &  \\\\
> \hline \text{CROWN-IBP CIFAR-10 }  \epsilon_{test}=8/255 &  \text{N/A} &\kappa : 1\rightarrow 0.5: 69.55 & \kappa : 1\rightarrow 0.5: 68.10 \\\\
>  &    & \kappa : 1\rightarrow 0: 67.76 & \\\\
>  &  &    \kappa : 0\rightarrow 0: 66.94 & \text{(batch size = 256)} \\\\
>  &  &     \text{(batch size = 1024)} &  \\\\
> \hline  \end{array}
> $$
>
> $\textbf{4. wrong claims and typo}$:
> Thank you for pointing out this wrong claim of "incomplete verification" in the original manuscript, we’ve listed other categories under “incomplete verification”. We have also corrected Equation (4)(Equation 2 in the revision pdf) in the revision and highlighted the changes.
>
>
> $\textbf{5. summary}$
>
> We briefly summarized our above response to your comments below:
> * #1: Theory: we addressed your concern on the bijection assumption in Theorem 1 and added a more rigorous definition for distribution $P’$.
> * #2: Algorithm: We corrected the definition of margin, explained the hyper-parameter generality and clarified the experiment baselines.
> * #3: We explained the generalizability/limitation of the method and included it into the new revision.
> * #4: We corrected the wrong claims and typos following your suggestion.
>
>
> Ref[A]: Gowal, S., Dvijotham, K., Stanforth, R., Bunel, R., Qin, C., Uesato, J., ... & Kohli, P. (2018). On the effectiveness of interval bound propagation for training verifiably robust models. arXiv preprint arXiv:1810.12715.
>
> Ref[B]: Zhang, H., Chen, H., Xiao, C., Gowal, S., Stanforth, R., Li, B., ... & Hsieh, C. J. (2019, September). Towards Stable and Efficient Training of Verifiably Robust Neural Networks. In International Conference on Learning Representations.

---

> > ### Author Response · Authors · 2021-11-22
> > **Requesting feedback**
> >
> > Dear reviewer: Since November 22nd is approaching, we want to get back to you and see if you have any remaining concerns. We have improved our paper using the suggestion from you and other reviewers. Please do not hesitate to contact us if you need further clarification. We hope to address all your concerns and you will consider re-evaluating our work.

---

> > > ### Comment · Reviewer_2goi · 2021-11-24
> > > **Feedback**
> > >
> > > I thank the authors for the clarifications of their proposed methods, experiments and theorem. I agree that the manuscript improves a little bit to some degree.
> > >
> > > For the theoretical side, I am still confused, because the cases where a part of examples with the same label swap the support are still not consistent with the assumption in Theorem 1. Theorem 1 assumes the perturbation is a bijection swapping the support of different classes. In addition, I do not believe the adversarial perturbations can swap the support of example even they are from the same class. For example, if you calculate the pixel-wise difference between images in MNIST or CIFAR10, we can find the norm of the difference is well above the adversarial budget $\epsilon$.
> > >
> > > For the experimental side, I thank the authors for the detailed clarification and think this should be included in the paper to facilitate the practitioners. Regarding the performance, I think the improvement is a bit marginal, making it hard to conclude that the improvement is from Algorithm 1.
> > >
> > > To conclude, I think this paper is still below the bar of acceptance although it has improvement from the original manuscript. I would give the score of 5.
> > >
> > > I welcome the author to further clarify my concerns.

---

> > > > ### Author Response · Authors · 2021-11-24
> > > > **Author follow-up response**
> > > >
> > > > Thank you for the follow-up response.
> > > >
> > > > Our Theorem 1 is an extreme case with a strong adversary. The practical case will be considered as a mixture of Theorem 1 assumed cases + other robust cases correctly classified under perturbation. Therefore, the latter does not conflict with Theorem 1 assumption.
> > > >
> > > > We agree with your opinion that for individual example pairs, it is difficult to swap the clean image support because the pixel difference is greater than the adversarial $\epsilon$. While we are targeting the distribution of all the adversarial examples within the neural network classification zones, not the support of clean examples from the data set. It is possible that a well-classified clean example, under certain perturbation $\epsilon$, will cross the decision boundary and be categorized into another class. Therefore, some clean examples will swap its distribution with some perturbed examples in another class from the NN’s point of view.
> > > >
> > > > Regarding the experiments, we’ve included the hyper-parameter study and a brief explanation of the experiment baseline into the manuscript. Although it is difficult to beat the SOTA, we still improve it consistently by 1-2%.  Meanwhile practically our methods are generally applicable and introduce negligible additional computational time, therefore we believe it is worthwhile to publish it.
> > > >
> > > > Thanks again for your valuable comments, please let us know if you still have concerns and we are very happy to further clarify.

---

### Official Review · Reviewer_onQv · 2021-11-01

**Correctness:** 3
**Technical Novelty And Significance:** 3
**Empirical Novelty And Significance:** 3
**Recommendation:** 8
**Confidence:** 5

**Details Of Ethics Concerns:**

No ethics concerns

**Main Review:**

Strength

1. Improving certifiable training performance is an important but very challenging task. The improvements are not marginal over IBP and CROWN-IBP compared to most of the recent papers.
2. The designs of the two methods are novel and reasonable. I like the examples in Section 3 which clearly illustrate the insights.

Weakness

1. Comprehensive experiments following CROWN-IBP would be necessary to fully demonstrate the performance of the methods. For instance, more model architectures should be evaluated. Also, 0.1 for MNIST and 2/255 for CIFAR10 are important baselines as well.
2. It would also be helpful to perform ablation studies for individual methods on CROWN-IBP besides IBP.
3. Despite the good results, I would not expect the proposed methods have a strong future impact enlightening further performance improvements since it is highly customized for the current stage and specific task. A discussion on limitations and future directions would be a plus.

Questions and comments

1. I am curious if applying the new methods on tighter relaxation like K&W [A, B], CROWN, or CROWN-IBP alone for certifiable training would improve the results or not?
2. A similar insight for improving certifiable training (CROWN-IBP) is proposed in [C] even though the eventual goal is slightly different. It would be helpful to at least mention it in the text.
3. Any specific reason for using "certification-based robust training" in the title instead of "certifiable training" which is a much more commonly used term?
4. Typo: Our bound-based method method, backed up by ...

[A] Provable defenses against adversarial examples via the convex outer adversarial polytope, ICML 2018

[B] Scaling provable adversarial defenses, NeurIPS 2018

[C] Adaptive Verifiable Training Using Pairwise Class Similarity, AAAI 2021

**Summary Of The Paper:**

This paper proposes bound-based weighted loss and epsilon auto-tuning to improve the performance of certifiable training. The insights of the improvements are mainly borrowed from well-developed adversarial training while they are customized for certifiable training considering bound margins provided bound propagation methods. The experimental results clearly show the improvements from individuals and the combination of the two methods.

**Summary Of The Review:**

Overall, I think the current version is a borderline paper. The proposed methods are new and the improvements shown in the paper are not that marginal compared to most of the related recent papers. However, the paper lacks comprehensive experiments to convincingly support the claims. If more consistent experimental results can be provided during rebuttal, I would vote for acceptance.

---

> ### Author Response · Authors · 2021-11-19
> **Author response 1/2**
>
> We are grateful for your positive feedback and insightful suggestions. We have $\textbf{conducted the suggested experiments}$ and results are consistent.
>
> $\textbf{1. Experiments}$
>
> Following your suggestion, we’ve conducted more experiments for MNIST eps =0.1, CIFAR eps=2/255 and another architecture DM-small following the CROWN-IBP paper Ref[E]. Due to limited rebuttal time, we conducted all these experiments with IBP bound, as CROWN-IBP training is much more expensive(roughly 5-7 times of computational time compared with IBP, ~2 days for each CIFAR DM-Large case on our hardware) and the existing CROWN-IBP results are consistent with IBP experiments. The new IBP results are also consistent with the trend from the main paper.
>
>
> $\text{Table R1:DM-small, MNIST, IBP, $\epsilon_{test}=0.3$}$
>
> $$
> \begin{array} {|r|r|r|r|r|}
> \hline \epsilon_{train}  & \text{re-weight} & \text{auto-eps}& \text{clean err.($\\%$) } & \text{verified err.($\\%$)  }  \\\\
> \hline 0.4 &  & & 3.27\pm0.24 &  12.00\pm0.35 \\\\
> \hline 0.4 &  \checkmark & & 3.25\pm0.10 &  11.73\pm0.57 \\\\
> \hline 0.4 &  & \checkmark & 2.75\pm0.11 &  11.36\pm0.24 \\\\
> \hline 0.4 &  \checkmark & \checkmark  & 2.87\pm0.1 &  11.09\pm0.33 \\\\
> \hline  \end{array}
> $$
>
> $\text{Table R2: DM-small, CIFAR-10, IBP, $\epsilon_{test}=8/255$}$
>
> $$
> \begin{array} {|r|r|r|r|r|}
> \hline \epsilon_{train}  & \text{re-weight} & \text{auto-eps}& \text{clean err.($\\%$) } & \text{verified err.($\\%$)  }  \\\\
> \hline 8.8/255 &   & & 51.86 \pm 0.18 &  73.92\pm 0.17 \\\\
> \hline 14/255 &  & & 58.08\pm0.33 &  71.07\pm0.23 \\\\
> \hline 14/255 &  \checkmark & & 57.56\pm0.76 &  70.33\pm0.29 \\\\
> \hline 14/255 &  & \checkmark & 58.36\pm1.05 &  70.62\pm0.46 \\\\
> \hline 14/255 &  \checkmark & \checkmark  & 57.29\pm0.17 &  70.02\pm0.18 \\\\
> \hline  \end{array}
> $$
>
> $\text{Table R3: DM-large, MNIST, IBP, $\epsilon_{test}=0.1$}$
>
> $$
> \begin{array} {|r|r|r|r|r|}
> \hline \epsilon_{train}  & \text{re-weight} & \text{auto-eps}& \text{clean err.($\\%$) } & \text{verified err.($\\%$)  }  \\\\
> \hline 0.2 &  & & 1.06\pm0.07 &  3.08\pm0.15 \\\\
> \hline 0.2 &  \checkmark & & 1.03\pm0.01 &  3.06\pm0.02 \\\\
> \hline 0.2 &  & \checkmark & 1.14\pm0.07 &  2.86\pm0.09 \\\\
> \hline 0.2 &  \checkmark & \checkmark  & 1.03\pm0.04 &  2.92\pm0.07 \\\\
> \hline  \end{array}
> $$
>
> $\text{Table R4: DM-large, CIFAR-10, IBP, $\epsilon_{test}=2/255$}$
>
> $$
> \begin{array} {|r|r|r|r|r|}
> \hline \epsilon_{train}  & \text{re-weight} & \text{auto-eps}& \text{clean err.($\\%$) } & \text{verified err.($\\%$)  }  \\\\
> \hline 2/255 &   & & 33.89 \pm 0.42 &  58.80\pm 0.84 \\\\
> \hline 6/255 &  & & 44.32\pm0.27 &  52.93\pm0.42 \\\\
> \hline 6/255 &  \checkmark & & 44.01\pm0.46 &  52.38\pm0.10 \\\\
> \hline 6/255 &  & \checkmark & 43.60\pm0.18 &  52.57\pm0.30 \\\\
> \hline 6/255 &  \checkmark & \checkmark  & 42.94\pm0.64 &  52.17\pm0.17 \\\\
> \hline  \end{array}
> $$
>
> $\textbf{2. Application on CROWN-IBP and other methods}$
>
> Thank you for your suggestion, we would like to highlight that our proposed method on weighted training is complimentary to all existing certifiable training e.g. IBP, K&W, CROWN-IBP, etc. The models trained with CROWN-IBP bounds achieve STOA certified accuracy and test accuracy, but are also much more expensive to train than IBP. As a proof-of-concept, we have shown our methods work for CROWN-IBP and IBP, and the literature(Ref[D]) suggested that if we include the verification bounds (e.g. IBP, crown-ibp) into training, we could make the verification bounds very tight, and even better than the more complicated verifier such as CROWN or Regarding K&W[A, B]: please see Table 4 Ref[D] mentioned that training with IBP bound could result in tighter testing bound compared with K&W[A, B]. Therefore, there is no need to train with a more complicated verification bounds e.g. K&W. As a result, though our method can be applied to K&W, in this paper we use IBP to demonstrate our method is effective.

---

> ### Author Response · Authors · 2021-11-19
> **Author response 2/2**
>
> $\textbf{3. Future work impact}$
>
> Yes we do agree with your opinion that the methods are customized for current perturbation and specific tasks. While during the process of evaluating different hyper-parameters, we use the same optimal re-weighting hyperparameter across all the tasks. For the auto-eps tuning, the $\epsilon_{off,max}$ requires empirical adjustment(the actual training $\epsilon_{train}$ cannot be too smaller than desired $\epsilon_{test}$, therefore $\epsilon_{off,max}$ must not be too large). We proposed our method and qualitatively explained the mechanism in the main paper. Unfortunately due to the intractable importance weight/optimal $\epsilon_{train}$, we need to parameterize a target function instead using the exact optimal. Once future work delivers more understanding of the adversarial mechanism, then it may be incorporated into our work and promotes the impact of this work in more general applications. We have included this discussion in the summary section of the revised paper and highlights the changes.
>
>
> $\textbf{4. other comments}$
>
> * Reference [C]: Thank you for pointing out this recent paper[C], we added it as discussion in the related work.
> * certification-based robust training: we don’t any preference over both of them, since “certifiable robust training” is more commonly used, we will ask the AC and update the title if possible.
>
> $\textbf{5. summary}$
>
> We briefly summarized our above response to your comments below:
> * #1: We conducted the suggested experiments and results are consistent with the main paper.
> * #2: We clarified our methods are applicable to CROWN-IBP.
> * #3: We explained the generalizability/limitation of the method and included it into the new revision.
> * #4: We modified the manuscript following your comments.
>
>
> Ref[A] Provable defenses against adversarial examples via the convex outer adversarial polytope, ICML 2018
>
> Ref[B] Scaling provable adversarial defenses, NeurIPS 2018
>
> Ref[C] Adaptive Verifiable Training Using Pairwise Class Similarity, AAAI 2021
>
> Ref[D]:
> Gowal, S., Dvijotham, K., Stanforth, R., Bunel, R., Qin, C., Uesato, J., ... & Kohli, P. (2018). On the effectiveness of interval bound propagation for training verifiably robust models. arXiv preprint arXiv:1810.12715.
>
> Ref[E]: Zhang, H., Chen, H., Xiao, C., Gowal, S., Stanforth, R., Li, B., ... & Hsieh, C. J. (2019, September). Towards Stable and Efficient Training of Verifiably Robust Neural Networks. In International Conference on Learning Representations.

---

> > ### Author Response · Authors · 2021-11-22
> > **Requesting feedback**
> >
> > Dear reviewer: Since November 22nd is approaching, we want to get back to you and see if you have any remaining concerns. We have improved our paper using the suggestion from you and other reviewers. Please do not hesitate to contact us if you need further clarification. We hope to address all your concerns and you will consider re-evaluating our work.

---

> > ### Comment · Reviewer_onQv · 2021-11-24
> > **Thanks for the response**
> >
> > Thank the authors for the detailed response and new experimental results. Most of my concerns have been addressed. There is one thing I want to clarify regarding the second point. The reason I am curious about certifiable training using tighter bound propagations is that greatly increasing tightness of the bound propagations could be one promising direction to further significantly improve certifiable training, even though the current versions of tighter ones available alone hardly end up with better results than weaker ones like IBP. it would be a strong plus for the future impact of the proposed method if it works well for tighter bound propagation certifiable training. However, I think this could be an interesting extension but not really necessary for this paper considering the current state of the arts. I am satisfied with the rest of the new experimental results. I will increase my score as promised.

---

> > > ### Author Response · Authors · 2021-11-24
> > > **Thank you for increasing the score**
> > >
> > > Dear reviewer,
> > >
> > > Thank you so much for updating the score. We really appreciate the valuable comments and discussions.
> > >
> > > Thanks again for bring up the tighter certifiable training bound, we are also curious if coupling with our methods could improve the testing bound. This will be one of the guidelines for our future work.
> > >
> > > All the best,
> > > Authors

---

### Official Review · Reviewer_p9xi · 2021-11-02

**Correctness:** 3
**Technical Novelty And Significance:** 2
**Empirical Novelty And Significance:** 2
**Recommendation:** 3
**Confidence:** 4

**Main Review:**

I think the weighting method is not presented that clearly in Section 3.1.
In particular, Equation 9 is quite confusing. Why can we just drop P'(x',y)/P(x,y) to get the final expression, isn't the whole point
that these two probabilities are quite different from each other (meaning the expression is not close to 1)?
Also how is final expression in Equation 9 any different from the standard loss from the beginning E_((x, y) ~ P) [l(f_\theta(x), y)]?
It seems to me that the only different is that the random variable has changed from x to x', but the overall expected loss should stay the same.

Final importance weighting that is used seems to be almost the same as the one used by Zeng et al., except that here we have
two hyperparameters alpha and gamma. Is this correct interpretation or there is some bigger difference between the two approaches?

Regarding auto-tuning of epsilon, prior work has already considered using larger epsilon to train the network,
so the key contribution here seems to be the algorithm presented in the Appendix C. As it is one of the key contributions of this work, it would make sense to put it
in the main paper, and not the appendix.

Final results seem somewhat underwhelming.
On MNIST the method improves 0.32% over the baseline, and on CIFAR-10 the improvement is 1.38%.
This seems quite small, but might be in the line with recent work in this field.
One question I had here is that your report the CROWN-IBP baseline to have verified error 68.35, but CROWN-IBP paper
reports 66.94. Could you explain where does the discrepancy come from?

Typos:

difition -> definition

examples with examples with -> examples with

the the ideal boundary -> the ideal boundary

linearly grows from 1 to 0.5 -> linearly decreases from 1 to 0.5


**Summary Of The Paper:**

This paper proposes two ideas for improving the performance of certified training.
The first idea is to use assign weight for each input based on the margin to the decision boundary.
The second idea is to use automatic scheduling of perturbation radius during training.
They show that using these two ideas leads to improved certified robustness on MNIST and CIFAR-10 datasets.


**Summary Of The Review:**

I recommend rejection because the method itself is incremental over prior work and experimental results are only marginally better.
Furthermore, I think paper would benefit from another revision as some things should be written more clearly.

---

> ### Author Response · Authors · 2021-11-19
> **Author response**
>
> Thank you for your careful review and the many constructive suggestions to clarify the paper. We believe that we have addressed all your main concerns, including $\textbf{clarifying the re-weighing method and experiment baseline discrepancy}$:
>
> $\textbf{1. Clarification for Section 3.1:}$
>
> The interpretation of Equation 9 (Equation 3 in the revision pdf file) is to correct the loss function from sampling over $x$ towards $x’$ by the importance weight. The second approximation could derived by following:
> $$
> E_{(x,y)\sim P}[l(f_\theta(x'),y)\frac{P'(x',y)}{P(x,y)}] =\sum_{(x,y)} P(x_i,y)[l(f_\theta(x'),y)\frac{P'(x',y)}{P(x,y)}] = \sum_{(x',y)} P(x_i^{\prime},y)l(f_\theta(x'),y) = E_{(x’,y)\sim P’}[l(f_\theta(x'),y)]
> $$
> $x'$ and $x$ have different prior so the overall expected loss $E_{(x’,y)\sim P}[l(f_\theta(x'),y)]$  and $E_{(x,y)\sim P}[l(f_\theta(x),y)]$ is not directly comparable. An extreme case is a very strong attack such that $l(f_\theta(x'),y)$ become very large for all the $x'$, $E_{(x,y)\sim P}[l(f_\theta(x),y)] $ is the clean loss which could be relatively small.
>
> $\textbf{2. Re-weight parametric function:}$
>
> In Ref[A],  the weight function monotonically decreases with the decision margin, which means the correctly classified labels get less attention compared with misclassified ones. This approach is based on intuitive understanding. Our proposed method focuses on the examples around the decision boundary due to the curvature, which we illustrated by examples. Although the format looks similar, it is one arbitrary pick of many ways to design a weight function growing/decreasing against the margin. More importantly, we have an absolute sign on the margin term in the equation making it behave completely differently from Ref[A].
>
> $\textbf{3. Appendix C}$
>
> Thank you for the suggestion, we've moved the Appendix C(auto-eps tuning) algorithm to the main paper.
>
> $\textbf{4. CROWN-IBP baseline}$
>
> Regarding the CROWN-IBP baseline, we explained the setup in the paper(p.6-7 Sec 4.1). Due to memory limitations of our hardware (Nvidia V100 with 32GB memory), we use a batch size of 256 instead of 1024 to run their released code directly. Note that the original CROWN-IBP paper is more resourceful and uses TPUv2 32 cores with 256GB memory. The other difference comes from the $\kappa$ schedule: the optimal in Ref[B] was achieved when $\kappa$ maintains 0, but we use a consistent $\kappa$ schedule(1->0.5) among all our experiments. Due to these two reasons, the baseline that we try to reproduced (68.10%) is higher than their reported optimal (66.94%) with $\kappa$ schedule(0->0.), while being better than their corresponding $\kappa$ schedule(1->0.5)(69.55%). To show the effect of auto-eps tuning, we elevated the $\epsilon_{train}$ to 14/255, therefore we have a (68.35%) as baseline. For a detailed baseline comparison, please refer to our reply to Reviewer $\textbf{2goi}$.
>
> $\textbf{5. typos}$
>
> Thank you for pointing out the typos, we have corrected them and highlighted the change in the new revision.
>
>
> $\textbf{6. summary}$
>
> We briefly summarized our above response to your comments below:
> * #1: We clarified your concern on Sec 3.1
> * #2: We explained the  the your concerns on the format of on re-weight function and its difference with Ref[A]
> * #3: We moved Appendix C to the main paper following your suggestion.
> * #4: We explained the CROWN-IBP baseline due to memory limited batch size.
> * #5: We corrected the typos.
>
>
>
> Ref[A]: Huimin  Zeng,  Chen  Zhu,  Tom  Goldstein,  and  Furong  Huang.    Are  adversarial  examples  cre-ated  equal?a  learnable  weighted  minimax  risk  for  robustness  under  non-uniform  attacks.arXiv:2010.12989, 2020.
>
> Ref[B]: Zhang, H., Chen, H., Xiao, C., Gowal, S., Stanforth, R., Li, B., ... & Hsieh, C. J. (2019, September). Towards Stable and Efficient Training of Verifiably Robust Neural Networks. In International Conference on Learning Representations.

---

> > ### Author Response · Authors · 2021-11-22
> > **Requesting feedback**
> >
> > Dear reviewer:
> > Since November 22nd is approaching, we want to get back to you and see if you have any remaining concerns. We have improved our paper using the suggestion from you and other reviewers. Please do not hesitate to contact us if you need further clarification. We hope to address all your concerns and you will consider re-evaluating our work.

---

> > > ### Comment · Reviewer_p9xi · 2021-11-22
> > > **Question on CROWN-IBP baseline**
> > >
> > > Thanks for your reply and updating the paper!
> > > I still have a concern with your CROWN-IBP baseline. First, note that CROWN-IBP github repo (https://github.com/huanzhang12/CROWN-IBP) states that the largest model can be trained in 1 day using 4 GPU-s which should certainly be doable.
> > > Second, I am quite concerned that you change $\kappa$ scheduling between baseline and your approach.
> > > CROWN-IBP achieves much better results using $1 \rightarrow 0$ and not $1 \rightarrow 0.5$ $\kappa$ schedule.
> > > Is it possible that your improvements (weighting and auto-tuning eps) are simply another way to achieve the same effect as $\kappa$ schedule achieves for CROWN-IBP?
> > >
> > > Given all this, it is difficult to judge how significant the proposed improvements are. I think proper experimental setup would be to take state-of-the-art CROWN-IBP model (which can apparently be trained without using TPU-s), use the same $\kappa$ scheduling and other parameters for both your model and the baseline, and then compare the results.

---

> > > > ### Author Response · Authors · 2021-11-22
> > > > **Author response**
> > > >
> > > > Thank you for the follow-up and we are glad to hear most of your concerns are addressed.
> > > >
> > > > Regarding computational speed, we conduct our experiments on a single Nvidia V100 graphic card so our 2-day computation time does not contradict CROWN-IBP github repo’s claim that the largest model could be trained within 1-day with 4 2080-Ti cards. While it is doable on our hardware, we still have to downsize the batch size from 1024 to 256 for CIFAR DM-large CROWN-IBP models.
> > > >
> > > > For the experiment baseline, as we mentioned above, our baseline is low for the largest CIFAR CROWN-IBP model due to batch size. We also noticed the optimal $\kappa$ schedule differs among different tasks and we choose the (1->0.5) because it consistently achieves best clean accuracy. But even with these 2 disadvantages, our optimal results(66.72%) are beating the CROWN-IBP paper(Ref[B])(66.94%) and in their repo(67.11%). For other setups, we’ve beating the SOTA by reaching 8.01% over 8.47% in MNIST IBP $\epsilon_{eval}=0.3$, 6.7% over 7.02% in MNIST CROWN-IBP $\epsilon_{eval}=0.3$, 68.33% over 70.81% in CIFAR IBP $\epsilon_{eval}=8/255$, where the former is our result with fix $\kappa$ epsilon and the later comes from the optimal $\kappa$ among 3 in CROWN-IBP paper(Ref[B]). Among all the experiments, our methods are consistently improving the baseline results regardless of using the task-dependent optimal $\kappa$. Overall, our methods are definitely not the other way to achieve a similar effect like choosing optimal $\kappa$.
> > > >
> > > > Please let us know if you need further clarification. We sincerely hope our explanation could clarify your concern and you will re-evaluate our paper.
> > > >
> > > > Ref[B]: Zhang, H., Chen, H., Xiao, C., Gowal, S., Stanforth, R., Li, B., ... & Hsieh, C. J. (2019, September). Towards Stable and Efficient Training of Verifiably Robust Neural Networks. In International Conference on Learning Representations.

---

> > > > > ### Comment · Reviewer_p9xi · 2021-11-25
> > > > > **Response: Baselines are still not properly evaluated**
> > > > >
> > > > > Thanks for your response. However, I still believe baselines are not properly evaluated.
> > > > > My concerns are listed below:
> > > > >
> > > > > 1. Given that this is the single most important result that we are interested in (certified robustness for CIFAR-10 with 8/255 perturbation), I think that it is reasonable to expect that you compare to the SOTA model given that computation time is relatively modest compared to some other deep learning models (1-2 days on 4 x 2080-Ti cards), and should definitely be achievable even in academic setting. Note that if large batch size is indeed the main problem for your hardware, you could also use virtual batching (e.g. you split batch size 1024 into 4 batches of 256 that you evaluate separately).
> > > > >
> > > > > 2. I don't find quite convincing your argument that $\kappa$-schedule 1->0.5 was chosen because it consistently achieves best clean accuracy. If our goal was to maximize clean accuracy, we would simply train with 1->1 (standard training). Instead, we need to train each method with hyperparameters for which the method achieves highest *certified robustness*.
> > > > >
> > > > > 3. You claim that your best model on CIFAR-10 with 66.72% beats CROWN-IBP which has 66.95%.
> > > > > I think that this improvement is both marginal and not statistically significant. Clearly, in your Table 5 we can
> > > > > see that the performance of your model is 66.72 +- 0.7 so there is quite a bit of variance around the actual certified robustness,
> > > > > and we cannot say with high confidence that your model is indeed better than the baseline.
> > > > >
> > > > > 4. You have added experiments with CIFAR-10 with 2/255 perturbation. However, here you again compare to weak baselines.
> > > > > For example, COLT [1] (not cited in your work) achieves clean error of 21.6% and verified error of 39.5% which is significantly better
> > > > > than the results you report in Table 6.
> > > > >
> > > > > [1] Balunovic, Mislav, and Martin Vechev. "Adversarial training and provable defenses: Bridging the gap." International Conference on Learning Representations. 2020.

---

> > > > > > ### Author Response · Authors · 2021-11-26
> > > > > > **author response to reviewer p9xi**
> > > > > >
> > > > > > Many thanks for your follow-up.
> > > > > >
> > > > > > Regarding the optimal $\kappa$ schedule, it varies among different experiments. We choose $\kappa$ 1->0.5 not only because it consistently achieves optimal clean accuracy, it also provides optimal robust accuracy for many tasks. For instance, half of MNIST tasks from Table 2,3 of Ref[B] achieve optimal robust accuracy with this setting. Therefore, we use a fixed hyper-parameter for both IBP and CROWN-IBP settings.
> > > > > >
> > > > > > We do agree with your concern that we need to run the optimal hyper-parameter for the baselines. For comprehensive study, we need to evaluate all the hyper-parameter options in the Ref[B] for reproducibility plus the combination of our methods. We do not think this is doable within the rebuttal window. From our perspective, the experiments are meant to show the general effect of our methods. Despite this specific task’s baseline under your concern, the results from our other experiments are consistent and convincing.
> > > > > >
> > > > > > Regarding your concern on the specific task of CROWN-IBP CIFAR-10 with 8/255 perturbation, we are trying to conduct some experiments under the optimal setting. Hopefully we can show part of the results by the discussion deadline and full results to the next revised version if the paper gets accepted.
> > > > > >
> > > > > > For the 2/255 baseline from paper COLT[1], it uses a different certification approach therefore it is not fair to bring it into direct comparison between our work and Ref[B]. COLT[1] also shows weaker performance under the 8/255 setting than Ref[B], thus we do not think it is necessary to show the application of our method on COLT[1]. What’s more important is that in our 2/255 experiments, we are using IBP in the training instead of CROWN-IBP. Therefore, our baseline(58.80%) should be compared with the IBP baseline(58.48%) in Ref[B] instead of CROWN-IBP.
> > > > > >
> > > > > > Thanks again for bringing up this concern. This is indeed valuable for our future work. We are trying to conduct some experiments. While your decision is critical and there is limited time for experiments, we hope you could understand our situation and re-evaluate our paper. If the paper gets accepted, we will include the experiments for this task in the camera-ready version.
> > > > > >
> > > > > > Ref[B]: Zhang, H., Chen, H., Xiao, C., Gowal, S., Stanforth, R., Li, B., ... & Hsieh, C. J. (2019, September). Towards Stable and Efficient Training of Verifiably Robust Neural Networks. In International Conference on Learning Representations.

---

> > > > > > ### Author Response · Authors · 2021-12-02
> > > > > > **author response to reviewer p9xi/updated results**
> > > > > >
> > > > > > Dear reviewer:
> > > > > >
> > > > > > Sorry for the late reply on updated results, we did our best to finish a set of results following your suggestions.
> > > > > >
> > > > > > We tried elevating the batch size from 256 to 1024 for our CROWN-IBP experiments with $\epsilon_{test}=8/255$, we achieve $66.08\pm0.38\\%$ as robust accuracy. In our run, the Ref[B]'s optimal hyper-parameter(batch size = 1024, $\kappa$  schedule 0->0)achieves $67.44\pm0.75\\%$ as they reported $66.94\\%$ in the paper. Both our experiments were conducted in 3 seeds.
> > > > > >
> > > > > > Sorry again for the delayed results due to limited time in the rebuttal. Our method beats the optimal baseline by $\sim 1\\%$ for this single most important task as you mentioned. We hope this addresses your concern on the baseline. Once the paper is accepted, we will include our experiments with a proper baseline for these specific tasks. Please consider to re-evaluate our paper if it is still possible for you.
> > > > > >
> > > > > > Best,
> > > > > >
> > > > > > Authors
> > > > > >
> > > > > > Ref[B]: Zhang, H., Chen, H., Xiao, C., Gowal, S., Stanforth, R., Li, B., ... & Hsieh, C. J. (2019, September). Towards Stable and Efficient Training of Verifiably Robust Neural Networks. In International Conference on Learning Representations.

---

### Author Response · Authors · 2021-11-19
**General response to all reviewers**

We would like to thank all the reviewers for their thoughtful comments. We appreciate the assessment of our paper as novel and insightful (Reviewer $\textbf{p9xi}$, $\textbf{onQv}$, $\textbf{jhkj}$) and clear motivation (Reviewer $\textbf{2goi}$). We also value constructive suggestions from all the reviewers.
Following all reviewers’ suggestions, we have revised the paper to include the following changes and highlighted the changes in red color in the revision manuscript:

$\bullet$ Move original appendix C (auto-eps tuning algorithm) to the main paper. (p.7)

$\bullet$ Add more related work and future work discussion. (p.1, p.9)

$\bullet$ Correct typos / improper claims.

$\bullet$ Conduct more experiments on different $\epsilon$ and other architectures. The updated results are included in the appendix. (p.15, table 6)

For some common questions, we would like to make two general replies:

$\bullet$Experiment baseline: our work was based on the code frame provided by Ref[A]. For each experiment, Ref[A] tries 3 different $\kappa$ schedules and optimal is among the 3. In our implementation, we use a fixed $\kappa$ schedule with optimal clean accuracy among all the experiments. Besides, for CROWN-IBP, we reduced the batch size from Ref[A]’s 1024 to 256 due to memory limitation. Therefore, our baseline is lower than but very close to Ref[A]’s optimal. For detailed comparison please refer to our reply to Reviewer $\textbf{2goi}$.

$\bullet$More experiments: following the reviewers’ suggestions, we conducted more experiments on DM-large CIFAR-10 $\epsilon_{test}=2/255$, MNIST $\epsilon_{test}=0.1$ and DM-small CIFAR-10 $\epsilon_{test}=8/255$, MNIST $\epsilon_{test}=0.3$. The performance of our methods are consistent with the experiments in the original manuscript. For details please refer to our reply to Reviewer $\textbf{onQv}$ and our revision manuscript.

For all the detailed replies, we replied under each reviewer’s comment. We kindly request you to read our clarifications and new additions to the paper. Any further discussions or clarification are very welcome!


Ref [A]: Zhang, H., Chen, H., Xiao, C., Gowal, S., Stanforth, R., Li, B., ... & Hsieh, C. J. (2019, September). Towards Stable and Efficient Training of Verifiably Robust Neural Networks. In International Conference on Learning Representations.

---

### Decision · Program_Chairs · 2022-01-20

**Decision:**

Reject

**Comment:**

The authors develop an approach to improve upon methods for training certifiably robust models. They propose an input dependent margin-based weighting and an automatically generated curriculum schedule and demonstrate improvements on training certifiably robust models on MNIST and CIFAR-10.

Reviewers agree that the paper makes interesting and novel contributions. However, the lack of novelty in the approach combined with the limited empirical gains make it difficult to justify acceptance. In particular, reviewers raise valid concerns on the quality of experiments comparing to prior work (in particular Crown-IBP (Zhang et al 2020) and COLT (Balunovic & Vechev 2020)) (in particular hyperparameter tuning, inability to recreate baseline results and unjustified claims that the prior art cannot run on GPU hardware). Further, even the gains demonstrated are marginal.

Hence, I recommend rejection, but encourage the authors to revise the paper based on the feedback received.